# 🌿 VISIONTS++: CROSS-MODAL TIME SERIES FOUNDATION MODEL WITH CONTINUAL PRE-TRAINED VISION BACKBONES

## ABSTRACT

Recent studies have indicated that vision models pre-trained on images can serve as time series foundation models (TSFMs) by reformulating time series forecasting (TSF) as image reconstruction. However, effective cross-modal transfer from vision to time series remains challenging due to three discrepancies: (1) the **data-modality gap** between structured, bounded image data and unbounded, heterogeneous time series; (2) the **multivariate-forecasting gap** between fixed RGB-three-channel vision models and time series with arbitrary numbers of variates; and (3) the **probabilistic-forecasting gap** between the deterministic outputs of vision models and the requirement for uncertainty-aware probabilistic predictions. To bridge these gaps, we propose VISIONTS++, a TSFM based on continual pre-training of a vision model on large-scale time series. Our approach introduces three key innovations: (1) **vision-model-based filtering** to identify high-quality sequences to stabilize pre-training and mitigate modality gap; (2) **colorized multivariate conversion**, encoding multivariate series as multi-subfigure RGB images to enhance cross-variate modeling; (3) **multi-quantile forecasting**, using parallel reconstruction heads to generate quantile forecasts without parametric assumptions. Experiments show that VISIONTS++ achieves state-of-the-art performance in both in-distribution and out-of-distribution forecasting, outperforming specialized TSFMs by 6%-44% in MSE reduction and ranking first in GIFT-Eval benchmark which comprises 23 datasets across 7 domains. Our work demonstrates that with appropriate adaptation, vision models can effectively generalize to TSF, thus advancing the pursuit of universal TSFMs. Code is available at https://anonymous.4open.science/r/VisionTSpp_1.

## 1 INTRODUCTION

Foundation models have transformed natural language processing (NLP) (Devlin et al., 2019; Radford et al., 2019) and computer vision (CV) (Dosovitskiy et al., 2021; He et al., 2022; Liu et al., 2021), motivating the development of *time series foundation models* (TSFMs) for *universal forecasting*—i.e., a single model that generalizes across diverse tasks without task-specific training (Woo et al., 2024; Ansari et al., 2024; Das et al., 2024; Shi et al., 2024). Yet, the heterogeneity of time series—spanning scale, frequency, and dimensionality—poses a major challenge to unified modeling (Liu et al., 2024c; Ansari et al., 2024; Liu et al., 2024b).

Recent work suggests that vision models pre-trained on images can be surprisingly effective for time series forecasting (TSF) (Shen et al., 2025; Ni et al., 2025; Xu et al., 2025). Notably, Chen et al. (2024b) shows that by reformulating univariate forecasting as image reconstruction, a Masked Autoencoder (MAE) pre-trained on natural images matches or exceeds specialized TSFMs. This hints at a conceptual alignment: images and time series may share similar patterns—*e.g.*, textures and edges in images can correspond to periodicities and trends in time series.

However, despite this promise, some fundamental discrepancies between them limit further improvements. Specifically, we identify three key gaps: **Data-Modality Gap**: Image pixels are bounded and spatially structured; while time series are unbounded and temporally heterogeneous. Directly applying vision models to TSF without appropriate adaptation is therefore suboptimal. **Multivariate-**

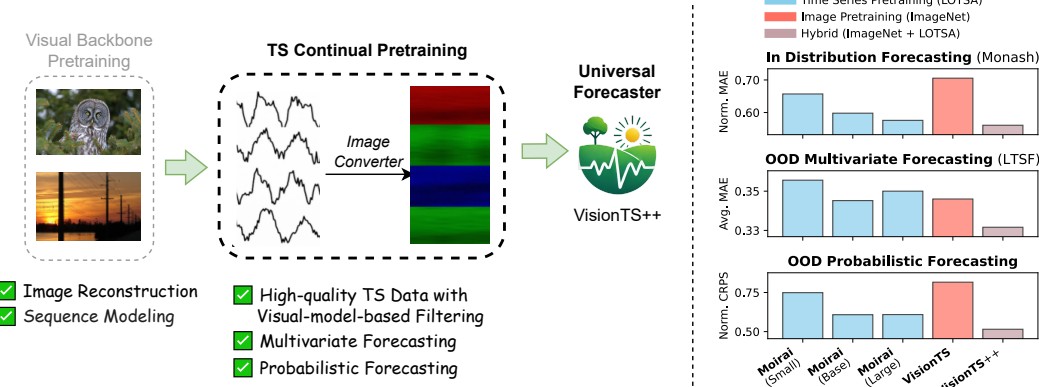

Figure 1: **Left**: Training pipeline of VISIONTS++. We perform continual pre-training of a visual backbone (`MAE`) on large-scale time series datasets to create a powerful and universal TSFM. **Right**: VISIONTS++ outperforms MOIRAI and VISIONTS in both multivariate and probabilistic forecasting, demonstrating its superior effectiveness.

**Forecasting Gap**: Vision models are designed with fixed RGB-three-channels, while multivariate time series have arbitrary numbers of variates (also referred to as *channels* in this paper). This hinders effective modeling of cross-variate dependencies. **Probabilistic-Forecasting Gap**: Most vision models focus on deterministic tasks like reconstruction, yet practical TSFMs require effective uncertainty-aware probabilistic predictions (Woo et al., 2024; Ansari et al., 2024; Liu et al., 2024b).

A straightforward yet blunt approach involves architectural modifications—*e.g.*, replacing input/output layers with time-series-specific modules (Zhou et al., 2023; Jin et al., 2023; Ansari et al., 2024)—followed by continual pre-training (CPT). However, such changes can disrupt valuable pre-trained visual representations, leading to negative transfer (Chen et al., 2024b; Wang et al., 2019; Liu et al., 2024a), and further degrade performance due to noisy or low-quality time series data (Liu et al., 2024c; Ansari et al., 2024). For example, Zhou et al. (2023) observe poor results when directly fine-tuning BeiT (Bao et al., 2021) for forecasting. This raises a critical question: **How can we effectively adapt a pre-trained vision model for TSF tasks, maximizing transfer effectiveness while robustly preserving its original knowledge?**

Building upon the framework of VISIONTS, our philosophy is to minimally modify the `MAE` architecture, and also transform TSF into image reconstruction. Based on this, we propose VISIONTS++, a vision-model-based TSFM that undergoes continual pre-training on large-scale time series, which supports flexible multivariate and probabilistic forecasting by efficiently transferring visual knowledge for TSF. Specifically, VISIONTS++ includes three key innovations to bridge the above gaps:

- **Vision-Model-Based Filtering**: To address the *data-modality gap*, we introduce a filtering mechanism that leverages the vision model itself to select high-quality time series. we identify and discard samples with out-of-range values or abrupt anomalies—inputs incompatible with the model's constraints. This enhances pre-training stability and mitigates negative transfer.

- **Colorized Multivariate Conversion**: To handle the *multivariate-forecasting gap*, we encode multivariate time series as multi-subfigure RGB images, where each variate is mapped to a distinct subfigure. This allows cross-variate dependencies to be better captured as spatial relationships between subfigures—naturally aligning with `MAE`'s multi-object analysis capability.

- **Multi-Quantile Forecasting**: To tackle the *probabilistic-forecasting gap*, we employ parallel reconstruction heads that generate multiple output images, each corresponding to a different quantile forecast. This reformulates probabilistic prediction as a set of deterministic image reconstructions—enabling flexible, assumption-free distribution modeling without relying on parametric priors (Woo et al., 2024).

After continual pre-training with these adaptations, VISIONTS++ achieves state-of-the-art (SOTA) performance across diverse forecasting tasks. For in-distribution forecasting, VISIONTS++ achieves the best normalized MAE on the Monash benchmark (Godahewa et al., 2021). For out-of-distribution

evaluations, VISIONTS++ outperforms existing TSFMs by 6%–44% in MSE reduction on the long-term TSF benchmark (Wu et al., 2021). It also ranks first in the Probabilistic Forecasting benchmark (Woo et al., 2024) and GIFT-Eval benchmark (Aksu et al., 2024) which comprises 23 datasets across 7 domains, beating many specialized TSFMs, thus demonstrating its strong generalization ability.

The training pipeline of VISIONTS++ is summarized in Figure 1. And our key contributions are summarized as follows:

- We propose VISIONTS++, a novel TSFM that performs continual pre-training of vision models on large-scale time series datasets, effectively adapting the model to time series temporal patterns while preserving pre-trained visual knowledge.

- We propose three targeted innovations—vision-model-based filtering, colorized multivariate time series conversion, and multi-quantile forecasting—that systematically address the data-modality, multivariate-forecasting, and probabilistic-forecasting gaps in cross-modal transfer.

- We demonstrate SOTA performance across in-distribution (*e.g.*, Monash) and out-of-distribution (*e.g.*, LTSF, PF, GIFT-Eval) benchmarks, establishing VISIONTS++ as a robust and general-purpose TSFM.

## 2 PRELIMINARIES

**Time Series Forecasting (TSF)**   For a multivariate time series with $M$ *variates* (also referred to as *channels* in this paper), let $\boldsymbol{x}_t \in \mathbb{R}^M$ represent the value at $t$-th time step. Then given a historical sequence (*i.e.*, look-back window) $\boldsymbol{X}_{t-L:t} = [\boldsymbol{x}_{t-L}, \cdots, \boldsymbol{x}_{t-1}] \in \mathbb{R}^{L \times M}$ with a context length of $L$, the TSF task is to use $\boldsymbol{X}_{t-L:t}$ to predict future values (*i.e.*, forecasting window): $\hat{\boldsymbol{X}}_{t:t+T} = [\hat{\boldsymbol{x}}_t, \cdots, \hat{\boldsymbol{x}}_{t+T-1}] \in \mathbb{R}^{T \times M}$, where $T$ is the prediction length.

**Image Reconstruction Task in `MAE`**   The Masked Autoencoder (`MAE`) (He et al., 2022) learns visual representations by reconstructing masked patches of an image. Given a square image of size $W \times W$, it is divided into $N \times N$ patches, each with a width and height of $S = {}^W\!/_N$. During pre-training, random patches are masked, and a Vision Transformer (ViT) (Dosovitskiy et al., 2021) is trained to reconstruct the missing pixel values based on the visible patches.

**Quick Review of VISIONTS**   Before introducing VISIONTS++, we briefly revisit the VISIONTS model (Chen et al., 2024b). Its core idea is to reformulate TSF as an image reconstruction task to adapt `MAE` for forecasting, which involves five key steps: (1) Segmentation and Image Conversion: It first segments a 1D time series $\boldsymbol{x} \in \mathbb{R}^L$ into periodic subsequences of length $P$, then arranges them into a 2D matrix $\boldsymbol{I}_{\text{raw}} \in \mathbb{R}^{P \times \lfloor L/P \rfloor}$. (2) Normalization and Rendering: After the instance normalization which yields $\boldsymbol{I}_{\text{norm}}$, the matrix is rendered into a grayscale-like image by repeating values across three RGB channels. (4) Alignment: To align with `MAE`'s input format, the image is resized to $(N \cdot S) \times (n \cdot S)$, where $n = \lfloor N \cdot L/(L+T) \rfloor$, so that the left portion corresponds to the context and the right portion (masked) to the forecast horizon. (5) Reconstruction and Time-series Conversion: The `MAE` model reconstructs the image, and the masked region is converted back to a 1D forecast through inverse operations.

## 3 METHODOLOGY

In this section we present VISIONTS++, a vision-model-based TSFM that adapts the pre-trained `MAE` backbone via **Continual Pre-training** (CPT) on large-scale time series data, enabling the vision model to align with the patterns of time series data. Building on VISIONTS (Chen et al., 2024b), we also reformulate TSF as an image reconstruction task. However, direct CPT is insufficient and faces three key challenges: the *Data-Modality Gap*, *Multivariate-Forecasting Gap*, and *Probabilistic-Forecasting Gap*, which hinder effective cross-modal transfer between images and time series. To bridge these gaps, we introduce three targeted designs—illustrated in Figure 2—that require minimal architectural changes while significantly improving adaptation and generalization.

## 3.1 VISION-MODEL-BASED FILTERING FOR TIME SERIES PRE-TRAINING

Firstly, to bridge the **Data-Modality Gap**, the core idea of VISIONTS++ is to perform continual pre-training (CPT) on large-scale time series data. However, the inherent heterogeneity and high noise in real-world time series raise concerns about data quality (Liu et al., 2024c; Ansari et al., 2024; Shi et al., 2024), thus demanding effective data curation approaches.

To obtain high-quality datasets, prior work in language models (Albalak et al., 2024; Goyal et al., 2024; Marion et al., 2023) and vision-language models (Chen et al., 2024c; Fang et al., 2023; Radenovic et al., 2023) has demonstrated that data filtering strategies can significantly improve dataset quality. Inspired by them, we explore the feasibility of similar techniques for time series—but a key question arises: "**How can we effectively filter low-quality time series to better bridge the data-modality gap for vision models?**"

To tackle this, we propose "**Vision-Model-Based Filtering**" (see bottom left part of Figure 2), which uses the vision model's own input constraints as a criterion to identify and filter out low-quality time series. This is based on the observation that vision models expect inputs within a bounded range (*e.g.*, image raw pixels in $[0, 255]$), whereas time series values are often unbounded. Time series containing out-of-range values can disrupt the model's pre-trained visual knowledge and harm transfer performance (Liu et al., 2024c; Ansari et al., 2024).

Specifically, pre-trained vision models expect inputs within a fixed range (between 0 and 255) derived from their training data (*e.g.*, ImageNet). Given pixel values, after normalization using dataset mean $\mu_I$ and standard deviation $\sigma_I$, valid inputs lie within the interval: $[(0-I_{\text{mean}})/I_{\text{std}}, (255-I_{\text{mean}})/I_{\text{std}}]$. Then for a time series input $\boldsymbol{X}_{t-L:t} \in \mathbb{R}^{L \times M}$ and target $\hat{\boldsymbol{X}}_{t:t+T} \in \mathbb{R}^{T \times M}$, we apply instance normalization using the context statistics $\mu_{\boldsymbol{X}} = \text{mean}(\boldsymbol{X}_{t-L:t})$ and $\sigma_{\boldsymbol{X}} = \text{std}(\boldsymbol{X}_{t-L:t})$.

Furthermore, to align the dynamic range with that of images, we follow VISIONTS and scale the normalized values by a factor $r = 0.4$, obtaining: $\boldsymbol{X}_{t-L:t}^{norm} = r \cdot \frac{\boldsymbol{X}_{t-L:t} - \mu_{\boldsymbol{X}}}{\sigma_{\boldsymbol{X}}}$ and $\boldsymbol{X}_{t:t+T}^{norm} = r \cdot \frac{\boldsymbol{X}_{t:t+T} - \mu_{\boldsymbol{X}}}{\sigma_{\boldsymbol{X}}}$ for both input and target. Despite this scaling, some values may still fall outside the valid visual input range. We thus filter out any sample for which $\boldsymbol{X}_{t-L:t}^{norm}$ or $\hat{\boldsymbol{X}}_{t:t+T}^{norm}$ contains values beyond $[(0-\mu_I)/\sigma_I, (255-\mu_I)/\sigma_I]$, ensuring compatibility with the vision model's input distribution.

## 3.2 COLORIZED MULTIVARIATE TIME SERIES CONVERSION

Having filtered high-quality samples, we need an image converter to transform multivariate time series into 2D images for the vision backbone. While VISIONTS (Chen et al., 2024b) processes each variate independently, this channel-wise isolation limits cross-variate modeling and increases computational overhead. A more scalable approach must support arbitrary numbers of variates within a unified visual representation. This leads to a critical question: **"How can we extend the image-based approach to better support efficient and effective multivariate time series forecasting?"**

A straightforward solution is to utilize the RGB channels as carriers for the multiple variates. However, there exists a significant "**Multivariate-Forecasting Gap**" between them: standard vision models assume exactly three input channels, which cannot naturally accommodate the high dimension of time series with arbitrary numbers of variates.

To bridge this gap, we propose "**Colorized Multivariate Conversion**", which treats each variate as a distinct **subfigure** within a single composite image (see top right of Figure 2). Rather than using RGB channels to encode variate values, we use them to define **spatial boundaries**, enabling the vision model to leverage its native multi-object analysis capability for cross-variate dependency modeling.

Formally, for input $\boldsymbol{X}_{t-L:t} \in \mathbb{R}^{L \times M}$, we follow VISIONTS to segment each variate into $\lfloor L/P \rfloor$ patches of length $P$ (periodicity), reshaping into a $P \times \lfloor L/P \rfloor$ matrix. This yields $\boldsymbol{I}_{\text{raw}} \in \mathbb{R}^{M \times P \times \lfloor L/P \rfloor}$. Each subfigure is then resampled to size $(\lfloor W/M \rfloor, W/2)$, where $W$ is the image width. Notably, we fix the visible and masked regions each to a width $W/2$, enabling efficient batch training across variable-length inputs.

Subsequently, $M$ subfigures are vertically stacked into a single image of size $(M \cdot \lfloor W/M \rfloor, W/2)$ and placed on the left side of the image. In case $M$ is not evenly divided by $W$, zero-padding is applied at the bottom of images. This layout ensures all variates are processed jointly in one forward pass.

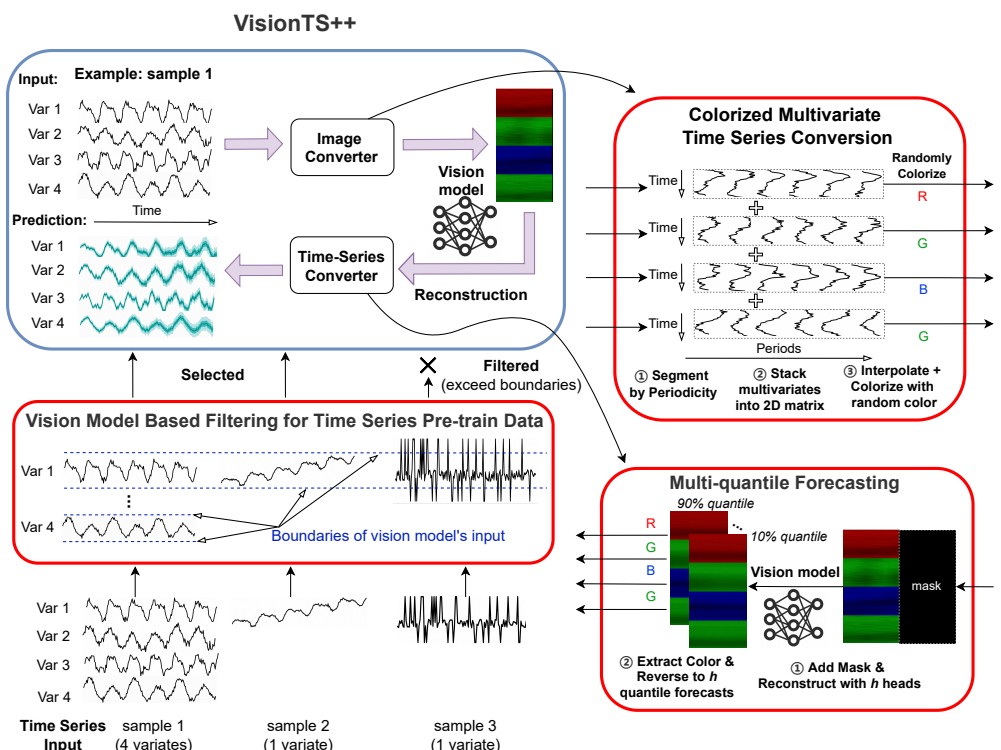

Figure 2: Overview of VISIONTS++. For each input, the following pipeline is applied: (1) Samples with out-of-range values after normalization are filtered out; (2) Each variate is segmented by periodicity and rendered as a colored subfigure, forming a composite image; (3) Multiple quantile forecasts are generated via parallel reconstruction heads. The model conducts continual pre-training on such transformed time series data to adapt `MAE` for universal forecasting.

Furthermore, to enhance clear boundaries between variates, we assign each subfigure a random RGB channel (others zeroed), with adjacent subfigures guaranteed to use different channels. This **color-as-boundary** strategy serves three important purposes: (1) activates the vision model's inherent multi-object capabilities; (2) prevents color bias through randomization; and (3) scales naturally to high-dimensional inputs.

### 3.3 MULTI-QUANTILE FORECASTING FOR PROBABILISTIC CONVERSION

After image conversion and masking, the visual backbone reconstructs the right half of the image. While this supports point forecasting, standard vision models that are designed for deterministic tasks lack inherent mechanisms for uncertainty quantification, which is a key requirement in most TSFMs (Woo et al., 2024; Ansari et al., 2024). We term such a limitation as the **Probabilistic-Forecasting Gap**. Therefore, this leads to a critical question: **"How can we transform the deterministically reconstructed image into meaningful probabilistic forecasts that accurately reflect the uncertainty in future time series?"**

To bridge this gap, we introduce the "**Multi-quantile Forecasting**" approach for the time series converter, which extends the vision model's native capabilities (See bottom right part of Figure 2). Instead of modeling distributions explicitly (*e.g.*, via parametric assumptions like Gaussian or Student's t (Flunkert et al., 2017; Woo et al., 2024) or via complex diffusion processes (Meijer & Chen, 2024; Li et al., 2022)), we approximate the full forecast distribution through multiple quantile estimates, each reconstructed as a separate image.

Specifically, we equip the vision model with $h$ parallel heads, each tasked with reconstructing the masked image region corresponding to a target quantile level $\tau_k = {k}/{h+1}$ for $k = 1, \ldots, h$.

Each head is trained with the quantile loss (See Section 3.4), enabling specialization across the distribution—covering tails and central regions alike.

During the image-to-time-series decoding, each reconstructed image is split vertically into $M$ subfigures. Values from the designated RGB channel are extracted, resampled from $(\lfloor W/M \rfloor, W/2)$ to $(P, \lfloor T/P \rfloor)$, and reassembled into a $(T, M)$-shaped time series. This yields $h$ quantile forecasts, forming a complete probabilistic output.

Notably, our approach offers several advantages: (1) It enables *seamless transfer learning*, repurposing pre-trained vision models for quantile forecasting with minimal architectural changes; (2) It performs *distribution-free uncertainty modeling*, avoiding restrictive assumptions about output distributions; (3) It supports *flexible quantile resolution*, allowing uncertainty granularity to be adjusted via the number of heads. Finally, the resulting framework thus unifies probabilistic and point forecasting: median quantiles (*e.g.*, $\tau = 0.5$) serve as robust point estimates, while the full set provides calibrated uncertainty intervals—making it adaptable to diverse downstream needs.

### 3.4 TRAINING OBJECTIVE

We train VISIONTS++ using a multi-quantile loss that jointly optimizes all $h$ forecasting heads. This objective supports probabilistic forecasting by supervising predictions across the full target distribution.

Specifically, let the target quantiles be $q_i = \frac{i}{h+1}$ for $i = 1, \ldots, h$, with corresponding forecasts $\boldsymbol{X}_{t:t+T}^{(i)}$ and ground truth $\hat{\boldsymbol{X}}_{t:t+T}$. The quantile loss (or pinball loss) for head $i$ is defined as:

$$\mathcal{L}_q = \frac{1}{h} \sum_{i=1}^{h} \max\left(q_i \cdot \boldsymbol{E}_i, \ (q_i - 1) \cdot \boldsymbol{E}_i\right), \quad \text{where} \quad \boldsymbol{E}_i = \hat{\boldsymbol{X}}_{t:t+T} - \boldsymbol{X}_{t:t+T}^{(i)}.$$

This loss ensures balanced optimization across quantiles, encouraging each head to specialize in its assigned level while sharing gradient signals across the ensemble. By avoiding point-estimate bias and making no distributional assumptions, it aligns naturally with our vision-based probabilistic framework and enables end-to-end training of the entire pipeline.

## 4 EXPERIMENTS

### 4.1 EXPERIMENTAL SETUP

**Training Dataset.** We conduct continual pre-training of VISIONTS++ on the Large-scale Open Time Series Archive (LOTSA) (Woo et al., 2024; Liu et al., 2024b), which is a diverse and multi-domain dataset containing over 231 billion observations. This scale and breadth can support robust temporal representation learning.

**Model Architecture.** We train two variants of VISIONTS++ of different scales (VISIONTS++$_{base}$ and VISIONTS++$_{large}$), based on the 112M and 330M parameter MAE (base) and MAE (large) architectures (He et al., 2022). Both are initialized from ImageNet pre-trained weights. Meanwhile, we set $h = 9$ quantile heads targeting levels $\{10\%, 20\%, \ldots, 90\%\}$ for probabilistic forecasting, balancing distributional coverage and model complexity.

**Training Process.** Continual pre-training runs for 100,000 steps with a batch size of 512. We use the AdamW optimizer (Loshchilov & Hutter, 2017) (learning rate: 1e−4, weight decay: 1e−2, momentum terms: $\beta_1 = 0.9$, $\beta_2 = 0.98$), with a learning rate schedule combining 10,000-step linear warm-up and subsequent cosine annealing. All model parameters are fine-tuned to fully adapt visual representations to TSF.

**Evaluation Protocol.** We follow recent TSFM research (Woo et al., 2024; Chen et al., 2024b; Liu et al., 2024b) and evaluate on three established benchmarks: Monash (Godahewa et al., 2021), Long-term Time Series Forecasting (LTSF) (Wu et al., 2021), and Probabilistic Forecasting (PF) (Woo et al., 2024), all compatible with LOTSA to avoid data leakage. We compare VISIONTS++ against state-of-the-art foundation models, deep learning, and classical baselines (details in Appendix C). Notably, our key comparisons include: (1) VISIONTS (ImageNet-pretrained) — to assess the impact

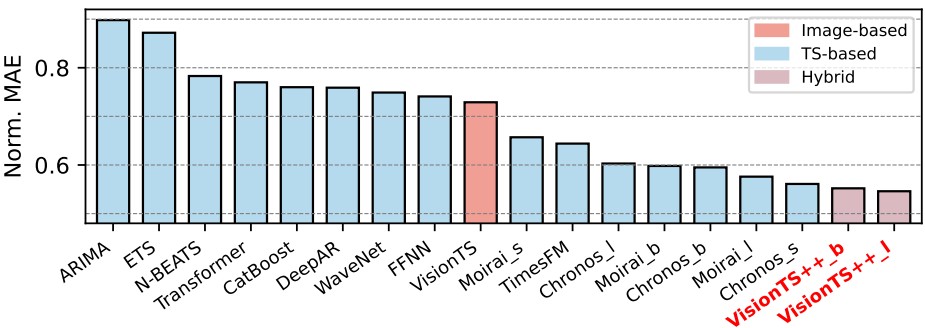

Figure 3: Normalized MAE results on Monash Benchmark, with full results in Table 4 (Appendix D.1). Model sizes are denoted as: **s** (small), **b** (base), **l** (large).

of CPT on temporal data adaptation; and (2) MOIRAI (LOTSA-pretrained) — to evaluate the benefit of visual pre-training. This dual comparison isolates the roles of modality transfer and temporal scaling in foundation models.

## 4.2 IN-DISTRIBUTION FORECASTING

**Monash Time Series Forecasting.** We evaluate in-distribution performance on a total of 29 datasets from the Monash benchmark (Godahewa et al., 2021) (details in Appendix C.1). To ensure a fair and rigorous comparison, the pre-training dataset LOTSA includes only the training portions of these series, with test sets held out for evaluation.

Figure 3 reports the normalized mean absolute error (nMAE), defined as the geometric mean of each model's MAE scaled by the naive forecasting baseline per dataset. The results show that VISIONTS++ achieves state-of-the-art performance across all models. It outperforms both dataset-specific models and the original VISIONTS by over 23.2%, validating the effectiveness of our conversion and pre-training paradigm. Notably, VISIONTS++ also surpasses MOIRAI —a foundation model trained on the same data—across all three sizes. This improvement, under identical training data and evaluation conditions, indicates that VISIONTS++'s ImageNet-pretrained visual knowledge provides a more effective initialization than training from scratch. The transferred visual representations enhance feature learning efficiency and in-distribution forecasting, demonstrating the value of cross-modal pre-training.

Table 1: Zero-shot results on LTSF benchmark of base and large models, averaged over four prediction lengths {96, 192, 336, 720}. Full results are in Table 5 (Section D.2). Time-MoE, Timer, and TimesFM are excluded in Electricity and Weather since time series were used in their pre-training.

| Dataset | Pre-train Method | 💰 Hybrid VisionTS++$_l$ | VisionTS++$_b$ | 🖼 Images VisionTS | Time-MoE$_s$ | Time-MoE$_b$ | 📈 Time-Series Chronos$_s$ | Chronos$_l$ | Moirai$_s$ | Moirai$_l$ | Moment | Timer$_{2sB}$ | TimesFM |
|---|---|---|---|---|---|---|---|---|---|---|---|---|---|
| ETTm1 | MSE | **0.354** | 0.360 | 0.374 | 0.394 | 0.376 | 0.640 | 0.556 | 0.448 | 0.390 | 0.670 | 0.487 | 0.433 |
| | MAE | **0.369** | 0.372 | 0.372 | 0.416 | 0.406 | 0.500 | 0.465 | 0.410 | 0.389 | 0.537 | 0.457 | 0.419 |
| ETTm2 | MSE | **0.244** | **0.244** | 0.282 | 0.318 | 0.316 | 0.349 | 0.295 | 0.300 | 0.276 | 0.317 | 0.316 | 0.328 |
| | MAE | **0.298** | **0.298** | 0.321 | 0.366 | 0.361 | 0.380 | 0.338 | 0.341 | 0.320 | 0.366 | 0.371 | 0.347 |
| ETTh1 | MSE | 0.403 | 0.402 | **0.390** | 0.400 | 0.394 | 0.545 | 0.589 | 0.400 | 0.510 | 0.684 | 0.444 | 0.473 |
| | MAE | 0.418 | 0.416 | **0.414** | 0.424 | 0.420 | 0.472 | 0.466 | 0.424 | 0.469 | 0.566 | 0.457 | 0.444 |
| ETTh2 | MSE | **0.327** | 0.333 | 0.333 | 0.367 | 0.405 | 0.424 | 0.455 | 0.341 | 0.354 | 0.362 | 0.358 | 0.392 |
| | MAE | **0.365** | 0.370 | 0.375 | 0.404 | 0.415 | 0.430 | 0.427 | 0.379 | 0.377 | 0.410 | 0.407 | 0.406 |
| Electricity | MSE | **0.181** | 0.184 | 0.207 | - | - | 0.220 | 0.204 | 0.233 | 0.188 | 0.765 | - | - |
| | MAE | **0.264** | 0.265 | 0.294 | - | - | 0.284 | 0.274 | 0.320 | 0.273 | 0.687 | - | - |
| Weather | MSE | 0.226 | **0.222** | 0.269 | 0.266 | 0.270 | 0.300 | 0.279 | 0.242 | 0.260 | 0.294 | 0.304 | - |
| | MAE | 0.243 | **0.241** | 0.292 | 0.297 | 0.300 | 0.318 | 0.306 | 0.267 | 0.275 | 0.326 | 0.331 | - |
| **Average** | MSE | **0.289** | 0.291 | 0.309 | - | - | 0.413 | 0.396 | 0.327 | 0.329 | 0.515 | - | - |
| | MAE | **0.326** | 0.327 | 0.345 | - | - | 0.397 | 0.379 | 0.357 | 0.350 | 0.482 | - | - |
| **1$^{st}$ count** | | 10 | 4 | 2 | 0 | 0 | 0 | 0 | 0 | 0 | 0 | 0 | 0 |

## 4.3 OUT-OF-DISTRIBUTION FORECASTING

To further evaluate the generalization capability, we conduct out-of-distribution forecasting (*i.e.*, zero-shot forecasting) experiments on two benchmarks—Long-term Time Series Forecasting (LTSF) (Wu et al., 2021) and Probabilistic Forecasting (PF) (Woo et al., 2024)—where neither training nor

test data overlap with the pre-training corpus LOTSA. This setup assesses the model's ability to transfer learned representations to unseen domains.

**Long-term Time Series Forecasting (LTSF).** We compare VISIONTS++ against state-of-the-art TSFMs including VISIONTS (Chen et al., 2024b), Time-MoE (Shi et al., 2024), Moirai (Woo et al., 2024), Chronos (Ansari et al., 2024), *etc.*Table 1 reports averaged Mean Squared Error (MSE) and Mean Absolute Error (MAE) across four prediction lengths $\{96, 192, 336, 720\}$ (full results in Table 5 in Appendix D.2).

The results show that VISIONTS++ achieves the best performance in 12 out of 14 settings. It improves over VISIONTS by 6% in average MSE, confirming that our image conversion and continual pre-training preserve visual priors while enhancing temporal modeling. Notably, VISIONTS++ outperforms specialized TSFMs by 6%–44% in MSE, demonstrating that with appropriate adaptation, vision-based models can surpass domain-specific architectures in long-term forecasting.

Table 2: Zero-shot results on the probabilistic forecasting benchmark. Best results are in **bold**.

| Dataset | Method | Zero-shot | | | | | | Full-shot | | | | Baseline | |
|---|---|---|---|---|---|---|---|---|---|---|---|---|---|
| | | VisionTS++$_l$ | VisionTS++$_b$ | VisionTS | Moirai$_s$ | Moirai$_b$ | Moirai$_l$ | PatchTST | TiDE | TFT | DeepAR | AutoARIMA | Seasonal Naive |
| Electricity | **CRPS** | **0.041** | 0.042 | 0.068 | 0.072 | 0.055 | 0.050 | 0.052±0.00 | 0.048±0.00 | 0.050±0.00 | 0.065±0.01 | 0.327 | 0.070 |
| | **MASE** | 0.635 | **0.631** | 0.755 | 0.981 | 0.792 | 0.751 | 0.753±0.01 | 0.706±0.02 | 0.747±0.03 | 0.844±0.16 | 3.229 | 0.881 |
| Solar | **CRPS** | **0.353** | 0.353 | 0.502 | 0.471 | 0.419 | 0.406 | 0.518±0.09 | 0.420±0.00 | 0.446±0.03 | 0.431±0.01 | 1.055 | 0.512 |
| | **MASE** | **1.135** | 1.155 | 1.141 | 1.465 | 1.292 | 1.237 | 1.607±0.25 | 1.265±0.02 | 1.399±0.11 | 1.222±0.01 | 2.583 | 1.203 |
| Walmart | **CRPS** | **0.061** | 0.064 | 0.121 | 0.103 | 0.093 | 0.098 | 0.082±0.01 | 0.077±0.00 | 0.087±0.00 | 0.121±0.00 | 0.124 | 0.151 |
| | **MASE** | **0.684** | 0.689 | 0.949 | 1.048 | 0.964 | 1.007 | 0.867±0.09 | 0.814±0.01 | 0.948±0.02 | 1.193±0.02 | 1.131 | 1.236 |
| Weather | **CRPS** | **0.038** | **0.038** | 0.056 | 0.049 | 0.041 | 0.051 | 0.059±0.01 | 0.054±0.00 | 0.043±0.00 | 0.132±0.11 | 0.252 | 0.068 |
| | **MASE** | 0.449 | **0.447** | 0.737 | 0.521 | 0.487 | 0.515 | 0.844±0.19 | 0.832±0.13 | 0.692±0.02 | 3.170±3.47 | 0.938 | 0.782 |
| Istanbul Traffic | **CRPS** | **0.105** | 0.115 | 0.198 | 0.173 | 0.116 | 0.112 | 0.112±0.00 | 0.110±0.01 | 0.110±0.01 | 0.108±0.00 | 0.589 | 0.257 |
| | **MASE** | **0.590** | 0.616 | 0.706 | 0.990 | 0.644 | 0.631 | 0.653±0.02 | 0.618±0.03 | 0.620±0.03 | 0.613±0.03 | 3.358 | 1.137 |
| Turkey Power | **CRPS** | 0.038 | **0.036** | 0.052 | 0.048 | 0.040 | **0.036** | 0.054±0.01 | 0.046±0.01 | 0.039±0.00 | 0.066±0.02 | 0.116 | 0.085 |
| | **MASE** | 0.752 | **0.737** | 0.856 | 0.948 | 0.888 | 0.870 | 1.234±0.12 | 0.904±0.02 | 0.890±0.05 | 1.395±0.30 | 1.700 | 0.906 |
| **Norm.** | CRPS | **0.506** | 0.515 | 0.816 | 0.749 | 0.608 | 0.609 | 0.679 | 0.612 | 0.595 | 0.857 | 2.123 | 1.000 |
| | MASE | **0.673** | 0.677 | 0.838 | 0.942 | 0.799 | 0.794 | 0.937 | 0.827 | 0.843 | 1.211 | 1.906 | 1.000 |
| **1st count** | | 10 | 6 | 0 | 0 | 0 | 1 | 0 | 0 | 0 | 0 | 0 | 0 |

**Probabilistic Forecasting (PF).** We further evaluate probabilistic forecasting on six real-world datasets (across energy, transport, climate, and sales domains) using the Continuous Ranked Probability Score (CRPS), along with MASE for point forecasting.

Based on results in Table 2, VISIONTS++ ranks first in all scenarios across both metrics. It significantly improves upon VISIONTS, validating the effectiveness of the multi-quantile forecasting design. More importantly, VISIONTS++ outperforms not only zero-shot but also full-shot baselines—despite receiving no dataset-specific training—highlighting its strong generalization. These results indicate that, with appropriate continual pre-training, vision-based TSFM can achieve SOTA zero-shot performance in probabilistic forecasting.

**GIFT-Eval Benchmark.** Additionally, we evaluate on the General Time Series Forecasting Model Evaluation (GIFT-Eval) benchmark (Aksu et al., 2024), which comprises 23 datasets across 7 domains. To ensure consistent evaluation, we re-train a version of VISIONTS++ using their "GiftEvalPretrain" dataset. strictly avoiding potential data leakage. We compare our model against TSFMs that similarly avoid data leakage, with baseline models cut-off as of the submission of VISIONTS++.

Based on results in Figure 4, VISIONTS++-large achieves the top rank under the aggregated ranking combining CRPS and MASE metrics, with the base model also ranking highly. Since GIFT-Eval includes both univariate and multivariate, as well as deterministic and probabilistic forecasting, this result demonstrates that VISIONTS++ effectively generalizes across diverse domains and supports a wide range of forecasting scenarios.

4.4 FURTHER ANALYSIS ON VISIONTS++

**Random Initialization.** To assess the importance of visual knowledge in the MAE backbone, we compare VISIONTS++ using ImageNet-pretrained weights versus random parameters for initialization before conducting continual pre-training. The results are reported in Table 6 in Appendix D.3 due to space limit.

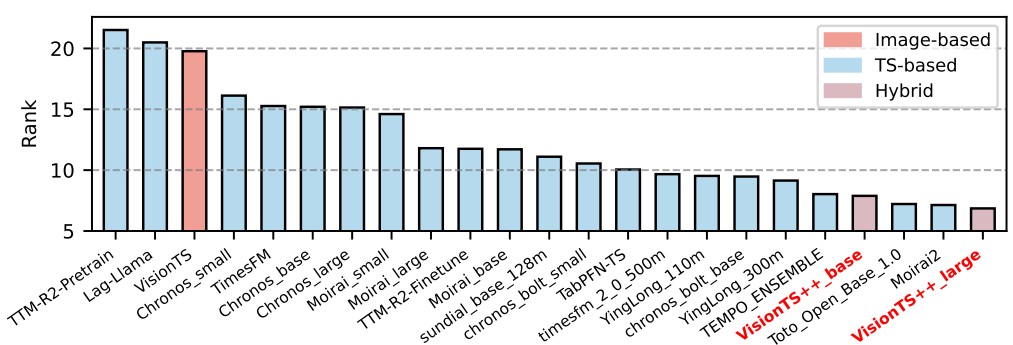

Figure 4: Ranks on GIFT-Eval Benchmark (cut-off at 2025/08).

These results reveal that the randomly initialized variant suffers a nearly 30% degradation in aggregated performance. This significant drop confirms that original visual representations provide an essential inductive bias for TSF, and that our continual pre-training effectively adapts—rather than overwrites—these features for time series.

**Computational Cost.** We evaluate the inference computational cost of different models on an NVIDIA A800 GPU, averaging over 100 runs. As shown in Table 3, VISIONTS++ achieves inference speed comparable to MOIRAI, VISIONTS, and GPT4TS, and is notably faster than TimesFM and LLMTime, which are autoregressive models. Specifically, compared to VISIONTS with a MAE (base) backbone (112M parameters), VISIONTS++ adds only eight additional forecasting heads, introducing just 3.0M extra parameters ($8 \cdot decoder\_dim \cdot patch\_size^2 \cdot n\_chans = 8 \cdot 512 \cdot 16^2 \cdot 3 = 3.0$M). This minimal parameter increase incurs negligible overhead in inference time.

Moreover, while most TSFM baselines exhibit increasing time with longer look-back window length $L$, VISIONTS++ maintains nearly constant latency. This is because VISIONTS++ encodes input sequences into a fixed-size image representation, achieving $O(1)$ inference efficiency versus the $O(L^2)$ complexity typical of Transformer-based approaches.

**Impact of Look-back Window Length.** Figure 5 shows the MSE performance of different look-back window lengths $L$ on the LTSF benchmark, averaged across four prediction horizons. We observe that VISIONTS++ 's performance generally improves as $L$ increases, but slightly degrades when $L$ becomes too large, which is consistent with other TSFMs (Moirai, VisionTS). This can be explained as: longer look-back windows initially provide richer historical context, improving forecasting accuracy. However, excessively large $L$ introduces two issues: (1) spatial constraints in a fixed-width image representation force interpolation that compresses or discards information, and (2) longer sequences may incorporate irrelevant or noisy context, slightly harming performance.

Table 3: Computational cost for forecasting a batch of 32 time series samples. (unit: second)

| Look-back Length | 1k | | | | 1k | 2k | 3k | 4k |
|---|---|---|---|---|---|---|---|---|
| Prediction Length | 1k | 2k | 3k | 4k | 1k | | | |
| PatchTST | 0.01 | 0.01 | 0.01 | 0.01 | 0.01 | 0.02 | 0.03 | 0.04 |
| DeepAR | 0.26 | 0.32 | 0.37 | 0.43 | 0.26 | 4.06 | 6.10 | 8.17 |
| LLMTime (8B) | > 200 | | | | > 200 | | | |
| Moirai$_{base}$ | 0.03 | 0.04 | 0.04 | 0.05 | 0.03 | 0.04 | 0.05 | 0.06 |
| TimesFM | 0.08 | 0.14 | 0.20 | 0.27 | 0.07 | 0.13 | 0.20 | 0.25 |
| VISIONTS | 0.04 | 0.03 | 0.03 | 0.03 | 0.04 | 0.04 | 0.05 | 0.05 |
| VISIONTS++ | 0.03 | 0.03 | 0.04 | 0.04 | 0.04 | 0.05 | 0.05 | 0.05 |

**Ablation Study.** We further ablate key components of VISIONTS++ (presented in Table 7) in Appendix D.3, demonstrating the contribution of each design:

- **Vision-model-based Filtering.** Removing this module leads to a 7% performance drop. It mitigates modality mismatch by filtering out extreme values that distort pixel-aligned visual representations, ensuring compatibility with the pre-trained backbone.

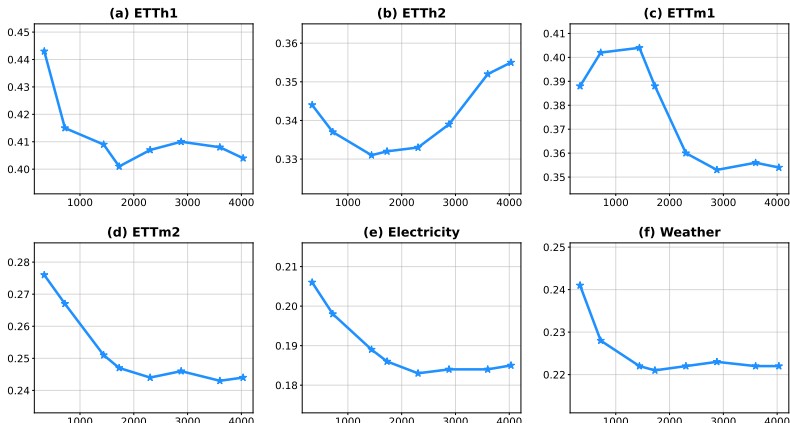

Figure 5: MSE performance of different look-back lengths $L$, averaged on four prediction lengths.

- **Colorized Multivariate Conversion.** Replacing RGB-encoded multivariate subfigures with grayscale univariate inputs (as in VISIONTS) increases MSE by 12%. The colorization strategy leverages the vision model's sensitivity to spatial and chromatic structure, enhancing cross-variate dependency modeling through multi-object analysis.

- **Multi-quantile Forecasting.** Collapsing to a single forecasting head degrades the probabilistic performance by over 10%. This validates that our unified design, which constructs and repurposes multiple MAE 's pixel reconstruction heads for quantile estimation, enables effective distributional forecasting.

## 5  CONCLUSIONS AND FUTURE WORK

In this paper, we propose VISIONTS++, a time series foundation model based on the continual pre-training of a vision foundation model on large-scale time series data. To bridge critical inherent gaps between images and time series, we introduce three key components, including vision-model-based filtering, colorized multivariate conversion, and multi-quantile forecasting. These designs enable effective adaptation of visual representations to time series patterns without modifying the underlying model architecture.

Extensive evaluation shows that VISIONTS++ achieves state-of-the-art performance across both in-distribution (Monash) and out-of-distribution (LTSF, PF) benchmarks, outperforming specialized TSFMs. These results demonstrate that pre-trained visual representations, when appropriately aligned with time series data, can serve as a powerful foundation for forecasting. Notably, our approach preserves valuable cross-modal knowledge while enabling robust temporal generalization—highlighting the potential of vision-based models in time series understanding.

Future work includes exploring larger-scale multi-modal pre-training, extending the framework to other time series tasks such as classification and anomaly detection, and investigating dynamic filtering mechanisms for diverse data regimes. Additionally, further integration with video foundation models may exploit spatio-temporal structure, advancing in more powerful universal models capable of unified visual and temporal understanding.

## ETHICS STATEMENT

This work poses no ethical concerns. The research does not involve human subjects, sensitive personal data, or potentially harmful applications. All datasets used are publicly available and were collected in compliance with their respective licenses and usage guidelines. Our model is designed for general-purpose time series forecasting and does not incorporate or amplify biases related to identity, race, gender, or other protected attributes. We adhere to the ICLR Code of Ethics and affirm that this work upholds principles of fairness, transparency, and research integrity.

REPRODUCIBILITY STATEMENT

We are committed to full reproducibility of our results. All implementation details, including model architectures, training procedures, hyperparameters, and data processing pipelines, are described in the main paper and Appendix. Our code, including training and evaluation scripts, will be released publicly upon acceptance. All datasets used are either publicly available or constructed from open sources with full documentation.

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

## APPENDIX

## A    USE OF LARGE LANGUAGE MODELS (LLMs)

Large language models were used only for improving the grammar and language fluency of the paper. All research ideas, methodological designs, technical developments, experimental setups, and analyses were conceived and conducted entirely by the authors without any assistance from LLMs. The scientific content and intellectual contributions are fully attributable to the authors.

## B    RELATED WORKS

### B.1    TIME SERIES FOUNDATION MODELS

Recent advances in time series forecasting have seen the emergence of time series foundation models (TSFMs) as powerful zero-shot forecasting tools. Unlike traditional dataset-specific models (*e.g.*, PatchTST (Nie et al., 2022), TiDE (Das et al., 2023), FEDformer (Zhou et al., 2022)) that require training on target datasets, TSFMs leverage large-scale pre-training to achieve cross-domain generalization. These models are typically pre-trained on diverse real-world time series datasets across diverse domains (Goswami et al., 2024; Liu et al., 2024c; Das et al., 2024; Dong et al., 2024; Feng et al., 2024) or pre-trained on synthetic time series data (Liu et al., 2025b; Fu et al., 2024; Yang et al., 2024). Notable examples include Moirai (Woo et al., 2024), which assembles a data archive of 231 billion observations across nine domains to train encoder-based models of varying scales, demonstrating strong zero-shot capabilities. Other foundation models with mostly encoder-based or decoder-based architectures have shown similar success, including Chronos (Ansari et al., 2024), TimesFM (Das et al., 2024), Timer (Liu et al., 2024c), Moment (Goswami et al., 2024), and Time-MoE (Shi et al., 2024). However, developing an effective TSFM faces significant challenges due to the inherent heterogeneity and high noise in time series data, thus demanding the construction of high-quality training datasets.

### B.2    VISION MODELS FOR TIME SERIES ANALYSIS

The exploration of vision-model-based approaches for time series analysis has significantly progressed in recent years. Early works demonstrate that encoding time series as images enables effective application of convolutional neural networks (CNNs) for both classification (Wang & Oates, 2015a;b; Hatami et al., 2018) and forecasting tasks (Li et al., 2020; Sood et al., 2021; Semenoglou et al., 2023). More recent advances have started to leverage pre-trained visual foundation models or vision-language models for time series analysis. For instance, AST Gong et al. (2021) adopts DeiT (Touvron et al., 2021) for time series classification, and ViTST (Li et al., 2023) utilizes pre-trained vision transformers (ViTs) (Dosovitskiy et al., 2021) and swin transformers (Liu et al., 2021) to further explore this direction. Other works, such as Wimmer & Rekabsaz (2023) and Zhang et al. (2023), explore the use of vision-language models for feature extraction and textual description generation. Moreover, ViTime (Yang et al., 2024) generates synthetic time series data and converts them into line plots for pre-training vision models such as ViT. ImagenTime (Naiman et al., 2024) introduces a unified generative framework by transforming time series into images via invertible methods like delay embedding and STFT, enabling them to leverage advanced vision diffusion models for generation, interpolation, and extrapolation tasks. Several recent surveys (Ni et al., 2025; Jiang et al., 2025; Liu et al., 2025a; Xu et al., 2025) have also discussed the application of vision models or multi-modal approaches in time series analysis. For example, Vision4TS (Ni et al., 2025) summarizes crucial techniques including time-series-to-image transformation, image pre-processing, and modeling strategies for imaged time series. The most relevant approach is VISIONTS (Chen et al., 2024b), which reformulates time series forecasting as a patch-level image reconstruction task and leverages the visual `MAE` model as the backbone.

However, although these methods establish preliminary connections between visual and time series domains, they fail to sufficiently address some critical modality gaps. To the best of our knowledge, we are the first to propose a competitive TSFM through continual pretraining on vision backbones, thus better enhancing the transferability between two modalities.

## C    BENCHMARKS & BASELINES

### C.1    BENCHMARKS

**Monash Benchmark**    Following Woo et al. (2024), we tested 29 Monash datasets (Godahewa et al., 2021) using GluonTS (Alexandrov et al., 2020), including M1 Monthly, M3 Monthly, M3 Other, M4 Monthly, M4 Weekly, M4 Daily, M4 Hourly, Tourism Quarterly, Tourism Monthly, CIF 2016, Australian Electricity Demand, Bitcoin, Pedestrian Counts, Vehicle Trips, KDD Cup, Weather, NN5 Daily, NN5 Weekly, Carparts, FRED-MD, Traffic Hourly, Traffic Weekly, Rideshare, Hospital,

COVID Deaths, Temperature Rain, Sunspot, Saugeen River Flow, and US Births. Performance is assessed using Mean Absolute Error (MAE) metric.

**Probabilistic Forecasting Benchmark**  The Probabilistic Forecasting (PF) Benchmark (Woo et al., 2024) consists of 6 datasets across energy, transport, climate, and sales domains, including Electricity, Solar, Walmart, Weather, Istanbul Traffic, and Turkey Power. Performance is assessed using Continuous Ranked Probability Score (CRPS) and Mean Absolute Scaled Error (MASE) metrics.

**Long-Term TSF Benchmark**  We evaluate our model on 6 widely used long-term TSF datasets (Zhou et al., 2021; Wu et al., 2021), including ETTh1, ETTh2, ETTm1, ETTm2, Electricity, and Weather. Performance is assessed using Mean Squared Error (MSE) and Mean Absolute Error (MAE) metrics.

**GIFT-Eval Benchmark**  Aksu et al. (2024) introduces the General Time Series Forecasting Model Evaluation (GIFT-Eval), encompasses 23 datasets over 144,000 time series and 177 million data points, spanning 7 domains, 10 frequencies, multivariate inputs, and prediction lengths ranging from short to long-term forecasts.

Table 4: Full results of Monash Time Series Forecasting Benchmark. MAE is reported.

| | VisionTS++_s | VisionTS++_b | VisionTS (z.s.) | LLMTime (z.s.) | Moirai_s | Moirai_b | Moirai_l | Naive | SES | Theta | TBATS | ETS | (DHR-)ARIMA | PR | CatBoost | FFNN | DeepAR | N-BEATS | WaveNet | Transformer |
|---|---|---|---|---|---|---|---|---|---|---|---|---|---|---|---|---|---|---|---|---|
| M1 Monthly | 1919.97 | 1,846.05 | 1987.69 | 2562.84 | 2,082.26 | 2,068.63 | 1,983.18 | 2,707.75 | 2,259.04 | 2,166.18 | 2,237.50 | 1,905.28 | 2,080.13 | 2,088.25 | 2,052.32 | 2,162.58 | 1,860.81 | 1,820.37 | 2,184.42 | 2,723.88 |
| M3 Monthly | 591.44 | 581.68 | 737.93 | 877.97 | 713.41 | 658.17 | 664.03 | 837.14 | 743.41 | 623.71 | 630.59 | 626.46 | 654.8 | 692.97 | 732 | 692.48 | 728.81 | 648.6 | 699.3 | 798.38 |
| M3 Other | 180.99 | 186.13 | 315.85 | 300.3 | 263.54 | 198.62 | 202.41 | 278.43 | 277.83 | 215.35 | 189.42 | 194.98 | 193.02 | 234.43 | 318.13 | 240.17 | 247.56 | 221.85 | 245.29 | 239.24 |
| M4 Monthly | 533.16 | 533.74 | 666.54 | 728.27 | 597.6 | 592.09 | 584.36 | 671.27 | 625.24 | 563.58 | 589.52 | 582.6 | 575.36 | 596.19 | 611.69 | 612.52 | 615.22 | 578.48 | 655.51 | 780.47 |
| M4 Weekly | 281.76 | 280.88 | 404.23 | 518.44 | 339.76 | 328.08 | 301.52 | 347.99 | 336.82 | 333.32 | 296.15 | 335.66 | 321.61 | 293.21 | 364.65 | 338.37 | 351.78 | 277.73 | 359.46 | 378.89 |
| M4 Daily | 190.54 | 172.31 | 215.63 | 266.52 | 189.1 | 192.66 | 189.78 | 180.83 | 178.27 | 178.86 | 176.6 | 193.26 | 179.67 | 181.92 | 231.36 | 177.91 | 299.79 | 190.44 | 189.47 | 201.08 |
| M4 Hourly | 169.17 | 202.99 | 288.37 | 576.06 | 268.04 | 209.87 | 197.79 | 1,218.06 | 1,218.06 | 1,220.97 | 386.27 | 3,358.10 | 1,310.85 | 257.39 | 285.35 | 385.49 | 886.02 | 425.75 | 393.63 | 320.54 |
| Tourism Quarterly | 5823.41 | 6,055.50 | 12931.88 | 16918.86 | 18,352.44 | 17,196.86 | 15,820.02 | 15,845.10 | 15,014.19 | 7,656.49 | 9,972.42 | 8,925.52 | 10,475.47 | 9,092.58 | 10,267.97 | 8,981.04 | 9,511.37 | 8,640.56 | 9,137.12 | 9,521.67 |
| Tourism Monthly | 1667.65 | 2,065.71 | 2560.19 | 5608.61 | 3,569.85 | 2,862.06 | 2,688.55 | 5,636.83 | 5,302.10 | 2,069.96 | 2,940.08 | 2,004.51 | 2,536.77 | 2,187.28 | 2,537.04 | 2,022.21 | 1,871.69 | 2,003.02 | 2,095.13 | 2,146.98 |
| CIF 2016 | 5664485.37 | 549,318.73 | 570907.24 | 599313.8 | 655,888.58 | 539,222.03 | 695,156.92 | 578,596.53 | 581,875.97 | 714,818.58 | 855,578.40 | 642,421.42 | 469,059.49 | 563,205.57 | 603,551.30 | 1,495,923.44 | 3,200,418.00 | 679,034.80 | 5,998,224.62 | 4,057,973.00 |
| Aus. Elec. Demand | 180.99 | 226.31 | 237.44 | 760.81 | 266.57 | 201.39 | 177.68 | 659.6 | 659.6 | 665.04 | 370.74 | 1,282.99 | 1,045.92 | 247.18 | 241.77 | 258.76 | 302.41 | 213.83 | 227.5 | 231.45 |
| Bitcoin | 1.85E+18 | 1.81E+18 | 2.33E+18 | 1.74E+18 | 1.76E+18 | 1.62E+18 | 1.87E+18 | 7.78E+17 | 5.33E+18 | 5.33E+18 | 9.90E+17 | 1.10E+18 | 3.62E+18 | 6.66E+17 | 1.93E+18 | 1.45E+18 | 1.95E+18 | 1.06E+18 | 2.46E+18 | 2.61E+18 |
| Pedestrian Counts | 61.47 | 62.55 | 52.01 | 97.77 | 54.88 | 54.08 | 41.66 | 170.88 | 170.87 | 170.94 | 222.38 | 216.5 | 635.16 | 44.18 | 43.41 | 46.41 | 44.78 | 66.84 | 46.46 | 47.29 |
| Vehicle Trips | 20.67 | 19.98 | 22.08 | 31.48 | 24.46 | 23.17 | 21.85 | 31.42 | 29.98 | 30.76 | 21.21 | 30.95 | 30.07 | 27.24 | 22.61 | 22.93 | 22 | 28.16 | 24.15 | 28.01 |
| KDD cup | 38.75 | 38.89 | 38.16 | 42.72 | 39.81 | 38.66 | 39.09 | 42.13 | 42.04 | 42.06 | 39.2 | 44.88 | 52.2 | 36.85 | 34.82 | 37.16 | 48.98 | 49.1 | 37.08 | 44.46 |
| Weather | 1.73 | 1.73 | 2.06 | 2.17 | 1.96 | 1.8 | 1.75 | 2.36 | 2.24 | 2.51 | 2.3 | 2.35 | 2.45 | 8.17 | 2.51 | 2.09 | 2.02 | 2.34 | 2.29 | 2.03 |
| NN5 Daily | 3.51 | 3.41 | 3.51 | 7.1 | 5.37 | 4.26 | 3.77 | 8.26 | 6.63 | 3.8 | 3.7 | 3.72 | 4.41 | 5.47 | 4.22 | 4.06 | 3.94 | 4.92 | 3.97 | 4.16 |
| NN5 Weekly | 14.84 | 14.12 | 14.67 | 15.76 | 15.07 | 16.42 | 15.3 | 16.71 | 15.66 | 15.3 | 14.98 | 15.7 | 15.38 | 14.94 | 15.29 | 15.02 | 14.69 | 14.19 | 19.34 | 20.34 |
| Carparts | 0.44 | 0.43 | 0.58 | 0.44 | 0.53 | 0.47 | 0.49 | 0.65 | 0.55 | 0.53 | 0.58 | 0.56 | 0.56 | 0.41 | 0.53 | 0.39 | 0.39 | 0.98 | 0.4 | 0.39 |
| FRED-MD | 2722.75 | 2,347.09 | 1893.67 | 2804.64 | 2,568.48 | 2,679.29 | 2,792.55 | 2,825.67 | 2,798.22 | 3,492.84 | 1,989.97 | 2,041.42 | 2,957.11 | 8,921.94 | 2,475.68 | 2,339.57 | 4,264.36 | 2,557.80 | 2,508.40 | 4,666.04 |
| Traffic Hourly | 0.013 | 0.016 | 0.01 | 0.03 | 0.02 | 0.02 | 0.01 | 0.03 | 0.03 | 0.03 | 0.04 | 0.03 | 0.04 | 0.02 | 0.02 | 0.01 | 0.01 | 0.02 | 0.02 | 0.01 |
| Traffic Weekly | 1.08 | 1.07 | 1.14 | 1.15 | 1.17 | 1.14 | 1.13 | 1.19 | 1.12 | 1.13 | 1.17 | 1.14 | 1.22 | 1.13 | 1.17 | 1.15 | 1.18 | 1.11 | 1.2 | 1.42 |
| Rideshare | 1.37 | 1.36 | 5.92 | 6.28 | 1.35 | 1.39 | 1.29 | 6.29 | 6.29 | 7.62 | 6.45 | 6.29 | 3.37 | 6.3 | 6.07 | 6.59 | 6.28 | 5.55 | 2.75 | 6.29 |
| Hospital | 17.30 | 17.00 | 19.36 | 25.68 | 23 | 19.4 | 19.44 | 24.07 | 21.76 | 18.54 | 17.43 | 17.97 | 19.6 | 19.24 | 19.17 | 22.86 | 18.25 | 20.18 | 19.35 | 36.19 |
| COVID Deaths | 114.97 | 151.53 | 137.51 | 653.31 | 124.32 | 126.11 | 117.11 | 353.71 | 353.71 | 321.32 | 96.29 | 85.59 | 85.77 | 347.98 | 475.15 | 144.14 | 201.98 | 158.81 | 1,049.48 | 408.66 |
| Temperature Rain | 4.83 | 5.17 | 6.37 | 6.37 | 5.3 | 5.08 | 5.27 | 9.39 | 8.18 | 8.22 | 7.14 | 8.21 | 7.19 | 6.13 | 6.76 | 5.56 | 5.37 | 7.28 | 5.81 | 5.24 |
| Sunspot | 0.25 | 0.19 | 2.81 | 5.07 | 0.11 | 0.08 | 0.13 | 3.93 | 4.93 | 4.93 | 2.57 | 4.93 | 2.57 | 3.83 | 2.27 | 7.97 | 0.77 | 14.47 | 0.17 | 0.13 |
| Saugeen River Flow | 23.24 | 24.24 | 30.22 | 34.4 | 24.07 | 24.4 | 24.76 | 21.5 | 21.5 | 21.49 | 22.26 | 30.69 | 22.38 | 25.24 | 21.28 | 22.98 | 23.51 | 27.92 | 22.17 | 28.06 |
| US Births | 420.22 | 411.48 | 519.94 | 1374.99 | 872.51 | 624.3 | 476.5 | 1,152.67 | 1,192.20 | 586.93 | 399 | 419.73 | 526.33 | 574.93 | 441.7 | 557.87 | 424.93 | 422 | 504.4 | 452.87 |
| **Normalized MAE** | **0.544** | **0.553** | 0.729 | 1.041 | 0.657 | 0.598 | 0.576 | 1.000 | 1.028 | 0.927 | 0.758 | 0.872 | 0.898 | 0.785 | 0.760 | 0.741 | 0.759 | 0.783 | 0.749 | 0.770 |

## C.2  BASELINES

**Baselines**  We select multiple representative baselines for comparison, including various time series foundation models as well as other popular TSF baselines covering Transformer-based and MLP-based architectures. These baseline models selected for comparison are briefly described below:

1. **VisionTS** (Chen et al., 2024b) is a vision-model-based TSF foundation model which utilizes the visual masked autoencoder pre-trained on ImageNet as the backbone model, and reformulate TSF as a patch-level image reconstruction task to complete prediction.

2. **Moirai** (Woo et al., 2024) is an encoder-based TSF foundation model trained on the Large-scale Open Time Series Archive (LOTSA), with over 231B observations across nine domains. It has three variants: **small**, **base**, and **large**.

3. **Time-MoE** (Shi et al., 2024) comprises a family of decoder-only transformer models, which leverages a sparse mixture-of-experts (MoE) design by activating only a subset of networks for each prediction to reduce computational load and maintain high model capacity.

4. **Chronos** (Ansari et al., 2024) tokenizes time series values using scaling and quantization into a fixed vocabulary, and trains T5 family language models (20M to 710M parameters) on these tokenized time series via the cross-entropy loss.

5. **Moment** (Goswami et al., 2024) family models serve as a building block for diverse time series analysis tasks, are effective out-of-the-box, and are tunable using in-distribution and task-specific data to improve performance.

6. **Timer** (Liu et al., 2024c) is a decoder-based TSF foundation model exhibiting similar characteristics to LLMs, such as flexible context length and autoregressive generation, along with notable few-shot generalization, scalability, and task generality.

Table 5: Full results of zero-shot forecasting on the long-term TSF benchmark. **Bold**: the best result.

| Pre-train | Hybrid | Hybrid | Images | Time-series | Time-series | Time-series | Time-series | Time-series | Time-series | Time-series | Time-series | Time-series | Time-series | Time-series |
|---|---|---|---|---|---|---|---|---|---|---|---|---|---|---|
| Method / Metric (MSE MAE) | VisionTS++$_I$ | VisionTS++$_b$ | VisionTS | Time-MoE$_b$ | Time-MoE$_b$ | Chronos$_s$ | Chronos$_b$ | Chronos$_l$ | Moirai$_s$ | Moirai$_b$ | Moirai$_l$ | Moment | Timer(28B) | TimesFM |
| ETTm1 96 | **0.312 0.342** | 0.316 0.343 | 0.341 0.347 | 0.338 0.368 | **0.309** 0.357 | 0.511 0.423 | 0.454 0.408 | 0.457 0.403 | 0.404 0.383 | 0.335 0.360 | 0.353 0.363 | 0.654 0.527 | 0.420 0.418 | 0.361 0.370 |
| ETTm1 192 | **0.341 0.360** | 0.347 0.362 | 0.360 **0.360** | 0.353 0.388 | 0.346 0.381 | 0.618 0.485 | 0.567 0.477 | 0.530 0.450 | 0.435 0.402 | 0.366 0.379 | 0.399 0.395 | 0.662 0.532 | 0.467 0.445 | 0.414 0.405 |
| ETTm1 336 | **0.361** 0.375 | 0.368 0.379 | 0.377 **0.374** | 0.377 0.413 | 0.373 0.408 | 0.683 0.524 | 0.662 0.525 | 0.577 0.481 | 0.462 0.416 | 0.391 0.394 | 0.399 0.395 | 0.672 0.537 | 0.502 0.467 | 0.445 0.429 |
| ETTm1 720 | **0.401 0.400** | 0.408 0.405 | 0.416 0.405 | 0.504 0.493 | 0.475 0.477 | 0.748 0.566 | 0.900 0.591 | 0.660 0.526 | 0.490 0.437 | 0.434 0.419 | 0.432 0.417 | 0.692 0.551 | 0.558 0.499 | 0.512 0.471 |
| ETTm1 avg | **0.354 0.369** | 0.360 0.372 | 0.374 0.372 | 0.394 0.416 | 0.376 0.406 | 0.640 0.500 | 0.646 0.500 | 0.556 0.465 | 0.448 0.410 | 0.382 0.388 | 0.390 0.389 | 0.670 0.537 | 0.487 0.457 | 0.433 0.419 |
| ETTm2 96 | **0.167 0.245** | 0.169 0.248 | 0.228 0.282 | 0.201 0.291 | 0.197 0.286 | 0.209 0.291 | 0.199 0.274 | 0.197 0.271 | 0.205 0.282 | 0.195 0.269 | 0.189 0.260 | 0.260 0.335 | 0.247 0.324 | 0.202 0.270 |
| ETTm2 192 | 0.217 0.280 | **0.216 0.279** | 0.262 0.305 | 0.258 0.334 | 0.250 0.322 | 0.280 0.341 | 0.261 0.322 | 0.254 0.314 | 0.261 0.318 | 0.247 0.303 | 0.247 0.300 | 0.289 0.350 | 0.294 0.358 | 0.289 0.321 |
| ETTm2 336 | 0.261 0.311 | **0.260 0.308** | 0.293 0.328 | 0.324 0.373 | 0.337 0.375 | 0.354 0.390 | 0.326 0.366 | 0.313 0.353 | 0.319 0.355 | 0.291 0.333 | 0.295 0.334 | 0.324 0.369 | 0.335 0.385 | 0.360 0.366 |
| ETTm2 720 | **0.329 0.358** | 0.330 0.358 | 0.343 0.370 | 0.488 0.464 | 0.480 0.461 | 0.553 0.499 | 0.455 0.439 | 0.416 0.415 | 0.415 0.410 | 0.355 0.377 | 0.372 0.386 | 0.394 0.409 | 0.386 0.418 | 0.462 0.430 |
| ETTm2 avg | **0.244 0.298** | **0.244 0.298** | 0.318 0.366 | 0.316 0.361 | 0.349 0.380 | 0.310 0.350 | 0.295 0.338 | 0.300 0.341 | 0.272 0.321 | 0.276 0.320 | 0.317 0.366 | 0.316 0.371 | 0.328 0.347 |  |
| ETTh1 96 | 0.368 0.392 | 0.369 0.392 | 0.353 0.383 | 0.357 **0.381** | **0.350** 0.382 | 0.466 0.409 | 0.440 0.393 | 0.441 0.390 | 0.375 0.402 | 0.384 0.402 | 0.380 0.398 | 0.688 0.557 | 0.393 0.421 | 0.414 0.404 |
| ETTh1 192 | 0.401 0.412 | 0.399 0.412 | 0.392 0.410 | **0.384 0.404** | 0.388 0.412 | 0.530 0.450 | 0.492 0.426 | 0.502 0.424 | 0.399 0.419 | 0.425 0.429 | 0.440 0.434 | 0.688 0.560 | 0.434 0.447 | 0.465 0.434 |
| ETTh1 336 | 0.416 0.424 | 0.415 **0.421** | **0.407** 0.423 | 0.411 0.434 | 0.411 0.430 | 0.570 0.486 | 0.550 0.462 | 0.502 0.424 | 0.412 0.429 | 0.456 0.450 | 0.514 0.474 | 0.675 0.563 | 0.460 0.464 | 0.503 0.456 |
| ETTh1 720 | 0.425 0.446 | 0.424 0.437 | **0.406** 0.441 | 0.449 0.477 | 0.427 0.455 | 0.615 0.543 | 0.882 0.591 | 0.835 0.583 | 0.413 0.444 | 0.470 0.473 | 0.705 0.568 | 0.683 0.585 | 0.487 0.494 | 0.511 0.481 |
| ETTh1 avg | 0.403 0.418 | 0.402 0.416 | **0.390 0.414** | 0.400 0.424 | 0.394 0.420 | 0.545 0.472 | 0.591 0.468 | 0.589 0.466 | 0.400 0.424 | 0.434 0.439 | 0.510 0.469 | 0.684 0.566 | 0.444 0.457 | 0.473 0.444 |
| ETTh2 96 | **0.267 0.317** | 0.277 0.326 | 0.271 0.328 | 0.305 0.359 | 0.302 0.354 | 0.307 0.356 | 0.308 0.343 | 0.320 0.345 | 0.281 0.334 | 0.277 0.327 | 0.287 0.325 | 0.342 0.396 | 0.308 0.369 | 0.315 0.349 |
| ETTh2 192 | 0.329 **0.361** | 0.333 0.362 | **0.328** 0.367 | 0.351 0.386 | 0.364 0.385 | 0.376 0.401 | 0.384 0.392 | 0.406 0.399 | 0.340 0.373 | 0.340 0.374 | 0.347 0.367 | 0.354 0.402 | 0.348 0.398 | 0.388 0.395 |
| ETTh2 336 | 0.350 **0.380** | 0.350 0.384 | **0.345 0.381** | 0.391 0.418 | 0.417 0.425 | 0.408 0.431 | 0.429 0.430 | 0.492 0.453 | 0.362 0.393 | 0.371 0.401 | 0.377 0.393 | 0.356 0.407 | 0.366 0.414 | 0.422 0.427 |
| ETTh2 720 | **0.362 0.401** | 0.370 0.409 | 0.388 0.422 | 0.419 0.454 | 0.537 0.496 | 0.604 0.533 | 0.501 0.477 | 0.603 0.511 | 0.380 0.416 | 0.394 0.426 | 0.404 0.421 | 0.395 0.434 | 0.409 0.446 | 0.443 0.454 |
| ETTh2 avg | **0.327 0.365** | 0.333 0.370 | 0.333 0.375 | 0.367 0.404 | 0.405 0.415 | 0.424 0.430 | 0.406 0.411 | 0.455 0.427 | 0.341 0.379 | 0.346 0.382 | 0.354 0.377 | 0.362 0.410 | 0.358 0.407 | 0.392 0.406 |
| Electricity 96 | **0.147 0.233** | 0.152 0.237 | 0.177 0.266 | - - | - - | 0.157 0.234 | 0.154 0.231 | 0.152 **0.229** | 0.205 0.299 | 0.158 0.248 | 0.152 0.242 | 0.745 0.680 | - - | - - |
| Electricity 192 | **0.164 0.250** | 0.168 0.252 | 0.188 0.277 | - - | - - | 0.183 0.258 | 0.179 0.254 | 0.172 0.252 | 0.220 0.310 | 0.174 0.263 | 0.171 0.259 | 0.755 0.683 | - - | - - |
| Electricity 336 | **0.184 0.268** | 0.186 0.269 | 0.207 0.296 | - - | - - | 0.220 0.290 | 0.214 0.284 | 0.203 0.276 | 0.236 0.323 | 0.191 0.278 | 0.192 0.278 | 0.766 0.687 | - - | - - |
| Electricity 720 | 0.229 0.303 | **0.228 0.303** | 0.256 0.337 | - - | - - | 0.321 0.353 | 0.311 0.346 | 0.289 0.337 | 0.270 0.347 | 0.229 0.307 | 0.236 0.313 | 0.794 0.696 | - - | - - |
| Electricity avg | **0.181 0.264** | 0.184 0.265 | 0.207 0.294 | - - | - - | 0.220 0.284 | 0.215 0.279 | 0.204 0.274 | 0.233 0.320 | 0.188 0.274 | 0.188 0.273 | 0.765 0.687 | - - | - - |
| Weather 96 | 0.146 0.179 | **0.145 0.179** | 0.220 0.257 | 0.160 0.214 | 0.159 0.213 | 0.211 0.243 | 0.203 0.238 | 0.194 0.235 | 0.173 0.212 | 0.167 0.203 | 0.177 0.208 | 0.243 0.255 | 0.243 0.283 | - - |
| Weather 192 | 0.190 0.221 | **0.187 0.219** | 0.244 0.275 | 0.210 0.260 | 0.215 0.266 | 0.263 0.294 | 0.256 0.290 | 0.249 0.285 | 0.216 0.250 | 0.209 0.241 | 0.219 0.249 | 0.278 0.329 | 0.288 0.320 | - - |
| Weather 336 | 0.245 0.261 | **0.240 0.258** | 0.280 0.299 | 0.274 0.309 | 0.291 0.322 | 0.321 0.339 | 0.314 0.336 | 0.302 0.327 | 0.260 0.282 | 0.256 0.276 | 0.277 0.292 | 0.306 0.346 | 0.323 0.345 | - - |
| Weather 720 | 0.324 0.313 | **0.317 0.308** | 0.330 0.337 | 0.418 0.405 | 0.415 0.400 | 0.404 0.397 | 0.397 0.396 | 0.372 0.378 | 0.320 0.322 | 0.321 0.323 | 0.365 0.350 | 0.350 0.374 | 0.362 0.374 | - - |
| Weather avg | 0.226 0.243 | **0.222 0.241** | 0.269 0.292 | 0.266 0.297 | 0.270 0.300 | 0.300 0.318 | 0.300 0.315 | 0.372 0.309 | 0.242 0.267 | 0.238 0.261 | 0.260 0.275 | 0.294 0.326 | 0.304 0.331 | - - |
| Average | **0.289 0.326** | 0.291 0.327 | 0.309 0.345 | - - | - - | 0.413 0.397 | 0.410 0.387 | 0.396 0.379 | 0.327 0.357 | 0.310 0.344 | 0.329 0.350 | 0.515 0.482 | - - | - - |
| 1st Count | 31 | 20 | 9 | 3 | 2 | 0 | 0 | 1 | 0 | 0 | 0 | 0 | 0 | 0 |

7. **TimesFM** (Das et al., 2024) is a decoder-style TSF foundation model, using a large time-series corpus comprising both real-world and synthetic datasets.

8. **LLMTime** (Gruver et al., 2023) encodes time series data to a text sequence, supporting zero-shot forecasting.

9. **Sundial** (Liu et al., 2025c) is a family of large-scale time series foundation models trained on over 100B real-world series, supporting flexible forecasting and strong zero-shot performance.

10. **Time-VLM** (Siru et al., 2025) reformulates time series as visual tokens and leverages pre-trained vision-language models for cross-modal forecasting.

11. **Lag-LLaMA** (Rasul et al., 2023) adapts the LLaMA architecture to time series using lag-based representations and autoregressive modeling.

12. **TTM (Tiny Time Mixers)** (Ekambaram et al., 2024) are lightweight, linear-mixer-based models that achieve fast inference and competitive accuracy with under 1M parameters.

13. **TabPFN-TS** (Hoo et al., 2024) treats forecasting as tabular regression over lagged features, enabling gradient-free few-shot adaptation.

14. **YingLong** (Wang et al., 2025) uses output scaling and delayed chain-of-thought decoding to improve long-horizon forecast accuracy.

15. **TEMPO** (Cao et al., 2023) is a prompt-based generative transformer that conditions forecasting on natural language or structured prompts.

16. **Toto** (Cohen et al., 2024) is a latency-optimized Transformer for observability tasks, handling irregular sampling and short-to-medium horizons efficiently.

17. **PatchTST** (Nie et al., 2022) uses Transformer encoders with patching and channel independence techniques for improved predictions.

18. **TiDE** (Das et al., 2023) is an MLP-based encoder-decoder TSF model, which enjoys the simplicity and speed of linear models while also being able to handle covariates and non-linear dependencies.

19. **TFT** (Lim et al., 2021) is an attention-based architecture which combines high-performance multi-horizon forecasting with interpretable insights into temporal dynamics.

For the long-term TSF benchmark, we include VISIONTS and other time series foundation models' results from their individual original papers. For the Monash and PF benchmark, we include all results from both Moirai and VISIONTS. For the GIFT-Eval benchmark, results are obtained from official code repository.

Table 6: Random initialization (right) vs. Loading MAE pre-trained weights (left) before CPT.

| | | VisionTS++$_b$ | rand_init |
|---|---|---|---|
| **Monash** | MAE | **0.553** | 0.733 |
| **PF** | MASE | **0.677** | 0.814 |
| | CRPS | **0.515** | 0.627 |
| **ETTm1** | MSE | **0.360** | 0.387 |
| | MAE | **0.372** | 0.396 |
| **ETTm2** | MSE | **0.244** | 0.29 |
| | MAE | **0.298** | 0.337 |
| **ETTh1** | MSE | **0.402** | 0.447 |
| | MAE | **0.416** | 0.45 |
| **ETTh2** | MSE | **0.333** | 0.47 |
| | MAE | **0.370** | 0.439 |
| **Electricity** | MSE | **0.184** | 0.225 |
| | MAE | **0.265** | 0.298 |
| **Weather** | MSE | **0.222** | 0.233 |
| | MAE | **0.241** | 0.257 |

Table 7: Ablation studies on each component in the VI-SIONTS++.

| | | VisionTS++$_b$ | w/o quantile | w/o filter | w/o color |
|---|---|---|---|---|---|
| **Monash** | MAE | **0.553** | 0.593 | 0.578 | 0.634 |
| **PF** | MASE | **0.677** | 0.714 | 0.690 | 0.725 |
| | CRPS | **0.515** | 0.551 | 0.531 | 0.565 |
| **ETTm1** | MSE | **0.360** | 0.392 | 0.388 | 0.408 |
| | MAE | **0.372** | 0.401 | 0.397 | 0.419 |
| **ETTm2** | MSE | **0.244** | 0.278 | 0.270 | 0.302 |
| | MAE | **0.298** | 0.328 | 0.324 | 0.356 |
| **ETTh1** | MSE | **0.402** | 0.421 | 0.416 | 0.453 |
| | MAE | **0.416** | 0.438 | 0.425 | 0.464 |
| **ETTh2** | MSE | **0.333** | 0.355 | 0.336 | 0.376 |
| | MAE | **0.370** | 0.387 | 0.372 | 0.402 |
| **Electricity** | MSE | **0.184** | 0.208 | 0.189 | 0.215 |
| | MAE | **0.265** | 0.288 | 0.272 | 0.299 |
| **Weather** | MSE | **0.222** | 0.234 | 0.228 | 0.245 |
| | MAE | **0.241** | 0.259 | 0.249 | 0.271 |

## D    FULL EXPERIMENTAL RESULTS

### D.1    FULL RESULTS FOR IN-DISTRIBUTION MONASH BENCHMARK

Table 4 provides the full breakdown of results ffor the Monash benchmark, listing results for each dataset in Monash. Based on the table, VISIONTS++ not only obtains SOTA overall normalized MAE results, but also achieves the best results in the vast majority of cases.

### D.2    FULL RESULTS FOR OUT-OF-DISTRIBUTION LTSF BENCHMARK

Table 5 provides the full detailed results for the long-term time series forecasting experiments, listing results for each prediction length. From the results, we can see that VISIONTS++ achieves the best results in most cases (large: 31 out of 62, and base: 20 out of 62), outperforming VISIONTS (9 out of 62), Time-MoE (3 out of 62), and all other models.

### D.3    FULL RESULTS FOR RANDOM INITIALIZATION AND ABLATION STUDY

We report the results of random initialization of VISIONTS++ in Table 6, and the results of full ablation studies in Table 7, due to space constraints in the main text. Detailed analysis of these results are provided in Section 4.4 in the main text.

### D.4    FULL RESULTS WITH STANDARD DEVIATIONS ON LTSF AND PF BENCHMARKS

We present full results with standard deviations on the LTSF benchmark in Table 8 and on the PF benchmark in Table 9, for both base and large variants of VISIONTS++. All results are averaged over five independent runs.

## E    FURTHER EXPERIEMENTS

### E.1    IMPACT OF QUANTILE HEAD NUMBER

We first examine the impact of the number of quantile heads. Table 10 compares the performance of the base variant of VISIONTS++ with 1, 3, and 9 quantile heads. Increasing the number of heads consistently improves nearly all metrics. The gain is especially pronounced on the PF benchmark in terms of CRPS—a key metric for probabilistic forecasting—since the quantile head design is intended to reduce the "probabilistic forecasting gap".

Table 8: Full results (Mean ± Std) on the LTSF benchmark. Values are averaged over five runs.

| Method | | VisionTS++$_l$ | | VisionTS++$_b$ | |
|---|---|---|---|---|---|
| Metric | | MSE | MAE | MSE | MAE |
| ETTm1 | 96 | 0.312±0.001 | 0.342±0.001 | 0.316±0.001 | 0.343±0.001 |
| | 192 | 0.341±0.002 | 0.360±0.001 | 0.347±0.002 | 0.362±0.001 |
| | 336 | 0.361±0.002 | 0.375±0.002 | 0.388±0.002 | 0.379±0.001 |
| | 720 | 0.401±0.001 | 0.400±0.001 | 0.408±0.001 | 0.405±0.001 |
| ETTm2 | 96 | 0.167±0.001 | 0.245±0.001 | 0.169±0.001 | 0.248±0.001 |
| | 192 | 0.217±0.002 | 0.280±0.001 | 0.216±0.002 | 0.279±0.001 |
| | 336 | 0.261±0.002 | 0.311±0.001 | 0.260±0.002 | 0.308±0.001 |
| | 720 | 0.329±0.003 | 0.358±0.002 | 0.330±0.002 | 0.358±0.001 |
| ETTh1 | 96 | 0.368±0.001 | 0.392±0.001 | 0.369±0.001 | 0.392±0.001 |
| | 192 | 0.401±0.002 | 0.412±0.001 | 0.399±0.002 | 0.412±0.001 |
| | 336 | 0.416±0.003 | 0.424±0.002 | 0.415±0.002 | 0.421±0.002 |
| | 720 | 0.425±0.002 | 0.446±0.002 | 0.424±0.002 | 0.437±0.002 |
| ETTh2 | 96 | 0.267±0.002 | 0.327±0.001 | 0.277±0.002 | 0.326±0.001 |
| | 192 | 0.329±0.003 | 0.361±0.002 | 0.333±0.002 | 0.362±0.001 |
| | 336 | 0.350±0.003 | 0.38±0.002 | 0.350±0.002 | 0.380±0.002 |
| | 720 | 0.362±0.003 | 0.401±0.002 | 0.370±0.002 | 0.409±0.002 |
| Electricity | 96 | 0.147±0.000 | 0.233±0.000 | 0.152±0.000 | 0.237±0.000 |
| | 192 | 0.164±0.000 | 0.250±0.000 | 0.168±0.001 | 0.252±0.001 |
| | 336 | 0.184±0.001 | 0.268±0.001 | 0.186±0.001 | 0.269±0.001 |
| | 720 | 0.229±0.001 | 0.303±0.001 | 0.228±0.002 | 0.303±0.001 |
| Weather | 96 | 0.146±0.000 | 0.179±0.000 | 0.145±0.000 | 0.179±0.000 |
| | 192 | 0.190±0.001 | 0.221±0.001 | 0.187±0.001 | 0.219±0.001 |
| | 336 | 0.245±0.001 | 0.261±0.001 | 0.240±0.001 | 0.258±0.001 |
| | 720 | 0.324±0.001 | 0.313±0.001 | 0.317±0.001 | 0.308±0.001 |

Table 9: Full results (Mean ± Std) on the PF benchmark. Values are averaged over five runs.

| Dataset | Method | VisionTS++$_l$ | VisionTS++$_b$ |
|---|---|---|---|
| Electricity | CRPS | 0.041±0.00 | 0.042±0.00 |
| | MASE | 0.635±0.00 | 0.631±0.00 |
| Solar | CRPS | 0.353±0.00 | 0.355±0.00 |
| | MASE | 1.135±0.01 | 1.155±0.01 |
| Walmart | CRPS | 0.061±0.00 | 0.064±0.00 |
| | MASE | 0.684±0.01 | 0.689±0.01 |
| Weather | CRPS | 0.038±0.00 | 0.044±0.00 |
| | MASE | 0.458±0.02 | 0.447±0.02 |
| Istanbul Traffic | CRPS | 0.105±0.01 | 0.115±0.01 |
| | MASE | 0.59±0.02 | 0.616±0.02 |
| Turkey Power | CRPS | 0.038±0.00 | 0.036±0.00 |
| | MASE | 0.752±0.01 | 0.737±0.01 |

Similar to Table 3 in Section 4.4, we also report inference computational cost for different head configurations in Table 11. Based on an MAE (base) backbone with 112M parameters, VISIONTS++ with 3 heads adds only two extra forecasting heads, introducing just 0.75M additional parameters $(2 \cdot 512 \cdot 16^2 \cdot 3 = 0.75M)$; the 9-head variant adds eight extra heads, totaling 3.0M additional parameters. This minimal parameter overhead incurs negligible increase in inference time, but yields substantial performance gains.

Table 10: Effect of quantile head numbers (1, 3, 9) on forecasting. Results show consistent improvements—especially in CRPS on the PF benchmark—as the number of heads increases.

| Dataset | Metric | 9-heads | 3-heads | 1-head |
|---|---|---|---|---|
| Monash | MAE | **0.553** | 0.561 | 0.570 |
| PF | MASE | **0.677** | 0.697 | 0.721 |
|  | CRPS | **0.515** | 0.675 | 0.804 |
| ETTm1 | MSE | **0.360** | 0.362 | 0.364 |
|  | MAE | **0.372** | 0.375 | 0.374 |
| ETTm2 | MSE | **0.244** | 0.249 | 0.251 |
|  | MAE | **0.298** | 0.301 | 0.302 |
| ETTh1 | MSE | 0.402 | 0.404 | **0.398** |
|  | MAE | 0.416 | 0.416 | **0.413** |
| ETTh2 | MSE | **0.333** | 0.336 | 0.340 |
|  | MAE | **0.370** | 0.375 | 0.372 |
| Electricity | MSE | **0.184** | 0.192 | 0.198 |
|  | MAE | **0.265** | 0.279 | 0.285 |
| Weather | MSE | **0.222** | 0.227 | 0.228 |
|  | MAE | **0.241** | 0.247 | 0.248 |

Table 11: Computational cost for forecasting a batch of 32 time series samples, with varying numbers of quantile heads.

| Look-back Length | 1k | | | | 1k | 2k | 3k | 4k |
|---|---|---|---|---|---|---|---|---|
| **Prediction Length** | 1k | 2k | 3k | 4k | 1k | | | |
| VISIONTS | 0.04 | 0.03 | 0.03 | 0.03 | 0.04 | 0.04 | 0.05 | 0.05 |
| VisionTS++$_b$ (1-head) | 0.03 | 0.03 | 0.03 | 0.03 | 0.04 | 0.04 | 0.05 | 0.05 |
| VisionTS++$_b$ (3-heads) | 0.03 | 0.03 | 0.03 | 0.04 | 0.04 | 0.05 | 0.05 | 0.05 |
| VisionTS++$_b$ (9-heads) | 0.03 | 0.03 | 0.04 | 0.04 | 0.04 | 0.05 | 0.05 | 0.05 |

### E.2    IMPACT OF SUBFIGURE LAYOUT

We evaluate different subfigure layouts during image reconstruction, including the "left-to-right" layout used in this paper, as well as "top-to-bottom" and "right-to-left" alternatives.

Table 12 summarizes the results. We observe that the choice of layout or orientation has little impact on performance. This strongly suggests that the image reconstruction process is isotropic—*i.e.*, invariant to spatial arrangement or directional bias.

### E.3    IMPACT OF COLOR ASSIGNMENT ORDER DURING INFERENCE

We also investigate the impact of color assignment order during inference. Two strategies are considered: (1) randomly assigning distinct colors to variates while ensuring adjacent subfigures receive different colors, and (2) cyclically coloring variates in a fixed top-to-bottom order (*e.g.*, repeating the red–green–blue cycle, as visualized in Figure 7 (a)).

Table 13 shows that the choice of color ordering has negligible effect on performance. This is because colorization serves primarily to explicitly indicate that adjacent subfigures correspond to different variates. During training, we randomize both the input variate order and the mapping from variates to RGB channels. After extensive pre-training, the model learns that color assignments are orthogonal to variate semantics and thus avoids spurious correlations. As a result, predictions at inference time

Table 12: Robustness of VISIONTS++ to subfigure layout, indicating the isotropic image reconstruction behavior.

| Dataset | Metric | left → right | top → bottom | right → left |
|---|---|---|---|---|
| Monash | MAE | **0.553** | 0.555 | 0.554 |
| PF | MASE | 0.677 | **0.676** | 0.677 |
| | CRPS | 0.515 | **0.513** | 0.516 |
| ETTm1 | MSE | **0.360** | 0.362 | **0.360** |
| | MAE | **0.372** | 0.375 | 0.373 |
| ETTm2 | MSE | **0.244** | 0.248 | 0.246 |
| | MAE | **0.298** | 0.303 | 0.300 |
| ETTh1 | MSE | 0.402 | 0.401 | **0.399** |
| | MAE | 0.416 | 0.415 | **0.413** |
| ETTh2 | MSE | **0.333** | 0.336 | **0.333** |
| | MAE | **0.370** | 0.374 | 0.371 |
| Electricity | MSE | **0.184** | 0.186 | 0.188 |
| | MAE | **0.265** | 0.268 | 0.270 |
| Weather | MSE | 0.222 | 0.223 | **0.221** |
| | MAE | 0.241 | 0.243 | **0.240** |

Table 13: Robustness of VISIONTS++ to color assignment strategy during inference, confirming that the model treats color as an auxiliary, non-semantic cue.

| Dataset | Metric | sequential color | random color |
|---|---|---|---|
| Monash | MAE | **0.553** | 0.555 |
| PF | MASE | **0.677** | 0.677 |
| | CRPS | **0.515** | 0.516 |
| ETTm1 | MSE | **0.360** | 0.361 |
| | MAE | **0.372** | 0.372 |
| ETTm2 | MSE | **0.244** | 0.244 |
| | MAE | **0.298** | 0.298 |
| ETTh1 | MSE | 0.402 | **0.401** |
| | MAE | 0.416 | **0.415** |
| ETTh2 | MSE | **0.333** | 0.334 |
| | MAE | **0.370** | 0.372 |
| Electricity | MSE | **0.184** | 0.184 |
| | MAE | **0.265** | 0.265 |
| Weather | MSE | **0.222** | 0.223 |
| | MAE | **0.241** | 0.241 |

remain robust to the specific colorization scheme. For reproducibility, we consistently adopt the fixed cyclic coloring strategy throughout this paper.

### E.4 COMPARISON WITH SUNDIAL AND TIME-VLM BASELINES

We further compare VISIONTS++ against recent strong baselines on the LTSF benchmark—including Sundial (large) (Liu et al., 2025c) and Time-VLM (Siru et al., 2025)—as summarized in Table 14.

First, both our VISIONTS++ and Sundial operate in a zero-shot forecasting setting. VISIONTS++ consistently outperforms Sundial across overall metrics, including average MSE, MAE, and 1st Count. Notably, Sundial is pre-trained on *TimeBench*, a dataset that fully subsumes our LOTSA corpus and is substantially larger—LOTSA constitutes only 22.29% of TimeBench's training data. Despite using far less pre-training data, VISIONTS++ achieves superior performance, demonstrating its data efficiency and effective architecture design.

Second, we compare with Time-VLM under its reported few-shot setting. Remarkably, our zero-shot VISIONTS++ outperforms Time-VLM on all metrics—even though Time-VLM Time-VLM is trained on some task-specific examples before inference while we do not. This highlights the strong generalization capability of our approach.

### E.5 PERIODICITY SELECTION

Accurately estimating the periodicity $P$ of the input time series is crucial, as our model relies on this signal for effective representation. Following VISIONTS, $P$ can be derived via statistical methods such as Fast Fourier Transform (Wu et al., 2023; Chen et al., 2024a) or from domain knowledge—*e.g.*, the known sampling frequency (Godahewa et al., 2021; Alexandrov et al., 2020). In this paper, we adopt the latter approach and set $P$ based on the sampling frequency.

However, not all time series exhibit clear periodicity. In such cases, we can fall back to setting $P = 1$. To validate the reasonableness of this choice, we deliberately use $P = 1$ even on the LTSF benchmark—which mostly contains periodic data—and report results in Table 15. While performance degrades compared to using the true periodicity, the results remain competitive: for instance, our model with $P = 1$ achieves average MSE and MAE comparable to those of Moirai-large (Woo et al., 2024).

The effectiveness of $P = 1$ stems from two factors: (1) the horizontal axis of the reconstructed image still preserves the full temporal trend of the input sequence, even without explicit periodic structure;

Table 14: Comparison between VISIONTS++-large (zero-shot) with Sundial-large (zero-shot) and Time-VLM (few-shot) on the LTSF benchmark.

| Method | | VisionTS++$_l$ | | Sundial$_l$ | | Time-VLM (few-shot) | |
|---|---|---|---|---|---|---|---|
| Metric | | MSE | MAE | MSE | MAE | MSE | MAE |
| ETTm1 | 96 | 0.312 | 0.342 | **0.273** | **0.329** | 0.314 | 0.357 |
| | 192 | 0.341 | 0.360 | **0.312** | **0.357** | 0.343 | 0.373 |
| | 336 | 0.361 | **0.375** | **0.343** | 0.378 | 0.373 | 0.391 |
| | 720 | 0.401 | **0.400** | **0.397** | 0.413 | 0.425 | 0.420 |
| | avg | 0.354 | **0.369** | **0.331** | **0.369** | 0.364 | 0.385 |
| ETTm2 | 96 | **0.167** | **0.245** | 0.172 | 0.255 | 0.169 | 0.260 |
| | 192 | **0.217** | **0.280** | 0.227 | 0.296 | 0.224 | 0.298 |
| | 336 | **0.261** | **0.311** | 0.275 | 0.331 | 0.282 | 0.338 |
| | 720 | **0.329** | **0.358** | 0.343 | 0.378 | 0.375 | 0.397 |
| | avg | **0.244** | **0.299** | 0.254 | 0.315 | 0.263 | 0.323 |
| ETTh1 | 96 | 0.368 | 0.392 | **0.346** | **0.383** | 0.417 | 0.435 |
| | 192 | 0.401 | 0.412 | **0.386** | **0.410** | 0.450 | 0.458 |
| | 336 | 0.416 | **0.424** | **0.410** | 0.426 | 0.460 | 0.465 |
| | 720 | **0.425** | **0.446** | 0.438 | 0.459 | - | - |
| | avg | 0.403 | **0.419** | **0.395** | 0.420 | - | - |
| ETTh2 | 96 | **0.267** | **0.327** | 0.269 | 0.330 | 0.302 | 0.365 |
| | 192 | **0.329** | **0.361** | 0.335 | 0.373 | 0.361 | 0.406 |
| | 336 | **0.350** | **0.380** | 0.354 | 0.400 | 0.398 | 0.434 |
| | 720 | **0.362** | **0.401** | 0.389 | 0.443 | - | - |
| | avg | **0.327** | **0.367** | 0.337 | 0.387 | - | - |
| Electricity | 96 | 0.147 | 0.233 | **0.130** | **0.227** | 0.185 | 0.296 |
| | 192 | 0.164 | 0.250 | **0.150** | **0.247** | 0.216 | 0.263 |
| | 336 | 0.184 | **0.268** | **0.170** | **0.268** | 0.264 | 0.298 |
| | 720 | 0.229 | **0.303** | **0.214** | 0.307 | 0.327 | 0.342 |
| | avg | 0.181 | 0.264 | **0.166** | **0.262** | 0.248 | 0.300 |
| Weather | 96 | **0.146** | **0.179** | 0.157 | 0.208 | 0.176 | 0.231 |
| | 192 | **0.190** | **0.221** | 0.207 | 0.256 | 0.216 | 0.263 |
| | 336 | **0.245** | **0.261** | 0.259 | 0.295 | 0.264 | 0.298 |
| | 720 | **0.324** | **0.313** | 0.327 | 0.342 | 0.327 | 0.342 |
| | avg | **0.226** | **0.244** | 0.238 | 0.275 | 0.246 | 0.284 |
| Average | | **0.289** | **0.327** | 0.291 | 0.338 | - | - |
| 1st Count | | 41 | | 23 | | 0 | |

and (2) during continual pre-training, the model has encountered numerous aperiodic time series, enabling it to generalize robustly to such scenarios.

Table 15: Impact of periodicity setting on forecasting in the LTSF benchmark. Even when forced to use $P = 1$ on the periodic data, VISIONTS++ maintains competitive results.

| Dataset | Metric | VisionTS++$_b$ | P=1 | Moirai$_l$ |
|---|---|---|---|---|
| ETTm1 | MSE | **0.360** | 0.448 | 0.390 |
| | MAE | **0.372** | 0.406 | 0.389 |
| ETTm2 | MSE | **0.244** | 0.296 | 0.276 |
| | MAE | **0.298** | 0.333 | 0.320 |
| ETTh1 | MSE | **0.402** | 0.438 | 0.510 |
| | MAE | **0.416** | 0.444 | 0.469 |
| ETTh2 | MSE | **0.333** | 0.399 | 0.354 |
| | MAE | **0.370** | 0.411 | 0.377 |
| Electricity | MSE | **0.184** | 0.202 | 0.188 |
| | MAE | **0.265** | 0.291 | 0.273 |
| Weather | MSE | **0.222** | 0.244 | 0.260 |
| | MAE | **0.241** | 0.266 | 0.275 |
| Average | MSE | **0.291** | 0.338 | 0.330 |
| | MAE | **0.327** | 0.359 | 0.351 |

### E.6 COMPARISON WITH ALTERNATIVE FILTERING STRATEGIES

To validate the effectiveness of our vision-model-based filtering approach, we compare it against two alternative strategies: (1) outlier detection based on the Interquartile Range (IQR) (Vinutha et al., 2018), and (2) change-point detection using Bayesian Online Changepoint Detection (BOCD) Adams & MacKay (2007).

For the IQR method, we first compute the first (Q1) and third (Q3) quartiles of the input time series and derive the IQR as $IQR = Q3 - Q1$, which captures the spread of the central 50% of the data. We then define upper and lower bounds as $Q3 + 1.5 \cdot IQR$ and $Q1 - 1.5 \cdot IQR$, respectively, and flag any values outside this range as outliers. Samples containing such outliers are filtered out.

For BOCD, we set the significance threshold $\alpha = 0.8$ for the posterior probability to detect changepoints in the input sequence. Samples flagged with change-points are discarded.

Table 16 reports the performance under each filtering strategy. Both alternatives yield consistently worse results compared to our vision-based filtering. This demonstrates that, despite its simplicity, our method is highly effective. It further considers the original visual representations learned by the MAE backbone, and better addresses the "data-modality gap" between time series and images.

### E.7 COMPARISON WITH ALTERNATIVE VISUAL BACKBONES

In this work, we adopt the MAE (He et al., 2022) model as our visual backbone. For comparison, we also evaluate alternative backbones, including SimMIM (Xie et al., 2022) and BootMAE (Dong et al., 2022), with results reported in Table 17.

Both alternatives underperform MAE. This observation aligns with recent findings in the literature. For instance, Zhao et al. (2025) attribute SimMIM's weaker performance on time series to its use of window-based local attention, which assumes translation invariance—an assumption invalid for imaged time series where spatial position encodes temporal order. Similarly, Shen et al. (2025) note that MAE's ViT-based reconstruction decoder is better suited for pixel-level forecasting tasks than SimMIM's linear decoder, despite their similar encoder architectures.

As for BootMAE, it introduces a momentum encoder (an exponential moving average of the main MAE encoder). While beneficial in some vision tasks, this design may over-smooth periodic or trend components in time-series-derived images, inadvertently discarding informative temporal structures.

Table 16: Performance under different filtering strategies, including Vision-Model-Based, IQR (Interquartile Range) and BOCD (Bayesian Online Changepoint Detection).

| Dataset | Metric | Vision-Model-Based | IQR | BOCD |
|---------|--------|--------------------|-----|------|
| Monash | MAE | **0.553** | 0.571 | 0.588 |
| PF | MASE | **0.677** | 0.686 | 0.699 |
|  | CRPS | **0.515** | 0.527 | 0.541 |
| ETTm1 | MSE | **0.360** | 0.378 | 0.392 |
|  | MAE | **0.372** | 0.390 | 0.402 |
| ETTm2 | MSE | **0.244** | 0.256 | 0.273 |
|  | MAE | **0.298** | 0.312 | 0.325 |
| ETTh1 | MSE | **0.402** | 0.409 | 0.422 |
|  | MAE | **0.416** | 0.421 | 0.438 |
| ETTh2 | MSE | **0.333** | 0.336 | 0.336 |
|  | MAE | **0.370** | 0.372 | 0.371 |
| Electricity | MSE | **0.184** | 0.187 | 0.189 |
|  | MAE | **0.265** | 0.268 | 0.275 |
| Weather | MSE | **0.222** | 0.224 | 0.229 |
|  | MAE | **0.241** | 0.243 | 0.249 |

Nevertheless, models built on both SimMIM and BootMAE still outperform several strong TSFM baselines (*e.g.*, VisionTS and Moirai-large), underscoring the effectiveness of our overall framework—particularly the continual pre-training on diverse time series data and our three targeted innovations. This also suggests that the performance stems not solely from the choice of MAE, but from the inherent compatibility between two modalities.

Table 17: Performance using different visual backbones, including MAE, SimMIM, and BootMAE.

| Dataset | Metric | MAE | SimMIM | BootMAE | VisionTS | Moirai$_l$ |
|---------|--------|-----|--------|---------|----------|------------|
| Monash | MAE | **0.553** | 0.605 | 0.579 | 0.729 | 0.576 |
| PF | MASE | **0.677** | 0.719 | 0.693 | 0.816 | 0.794 |
|  | CRPS | **0.515** | 0.573 | 0.535 | 0.838 | 0.609 |
| ETTm1 | MSE | **0.360** | 0.401 | 0.380 | 0.374 | 0.390 |
|  | MAE | **0.372** | 0.403 | 0.393 | 0.372 | 0.389 |
| ETTm2 | MSE | **0.244** | 0.278 | 0.259 | 0.282 | 0.276 |
|  | MAE | **0.298** | 0.323 | 0.315 | 0.321 | 0.320 |
| ETTh1 | MSE | **0.402** | 0.428 | 0.413 | 0.390 | 0.510 |
|  | MAE | **0.416** | 0.443 | 0.424 | 0.414 | 0.469 |
| ETTh2 | MSE | **0.333** | 0.356 | 0.349 | 0.333 | 0.354 |
|  | MAE | **0.370** | 0.387 | 0.381 | 0.375 | 0.377 |
| Electricity | MSE | **0.184** | 0.205 | 0.192 | 0.207 | 0.188 |
|  | MAE | **0.265** | 0.288 | 0.279 | 0.294 | 0.273 |
| Weather | MSE | **0.222** | 0.239 | 0.229 | 0.269 | 0.260 |
|  | MAE | **0.241** | 0.260 | 0.250 | 0.292 | 0.275 |

# F  VISUALIZATION

## F.1  ATTENTION MAPS

We visualize the attention maps in Figure 6. Specifically, we sample one instance each from ETTh1 and ETTh2 datasets and plot the attention scores from the last decoder layer of our model. Given the `MAE` 's patch size of 16, the 224-pixel height is divided into 14 patches. Since both ETTh1 and ETTh2 contain 7 variates, each variate corresponds to exactly 2 consecutive patches vertically, and we visualize the split lines using red dashed lines.

In Figure 6 (a), the first token of variate 1 is used as the query. We can observe relatively high attention scores with the first tokens of variates 2, 3, and 7. Although these tokens are not adjacent in the input image, they align temporally since they represent different variates at the same time step. Similarly, in Figure 6 (b), when using the first token of variate 4 as the query, high attention scores are also observed for variates 2 and 3 at the same time step.

These examples demonstrate that, despite being a vision-based TSFM, VISIONTS++ can effectively capture cross-variate dependencies. This should be owing to thorough continual pre-training on time series data and well-designed components to reduce the modality gaps.

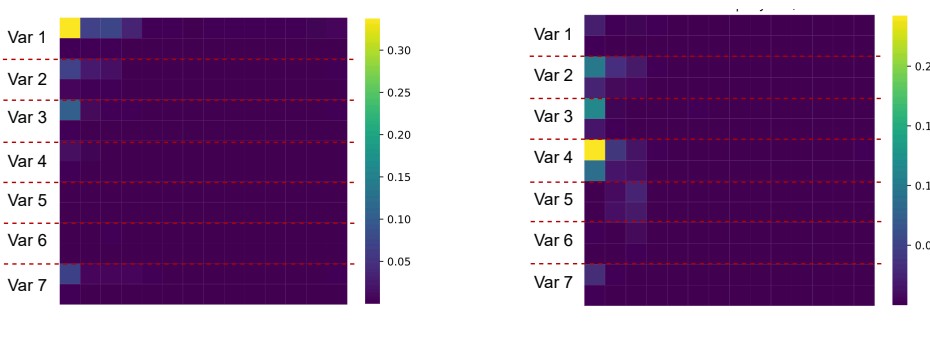

(a) Query: first token of variate 1 (ETTh1)          (b) Query: first token of variate 4 (ETTh2)

Figure 6: Attention maps from the last decoder layer of VISIONTS++, showing strong cross-variate attention at co-temporal positions.

## F.2   CASE PRESENTATION

In this section, we visualize the multivariate time series predictions of VISIONTS++ in the zero-shot setting, including its input and reconstructed images. We also visualize its predictions, with MSE and MAE metrics for comparison. These samples are presented in Figure 7 and Figure 8.

These examples show the superior forecasting performance of VISIONTS++ over VISIONTS after conducting the continual pre-training, as well as other components that effectively address the modality gaps between images and time series.

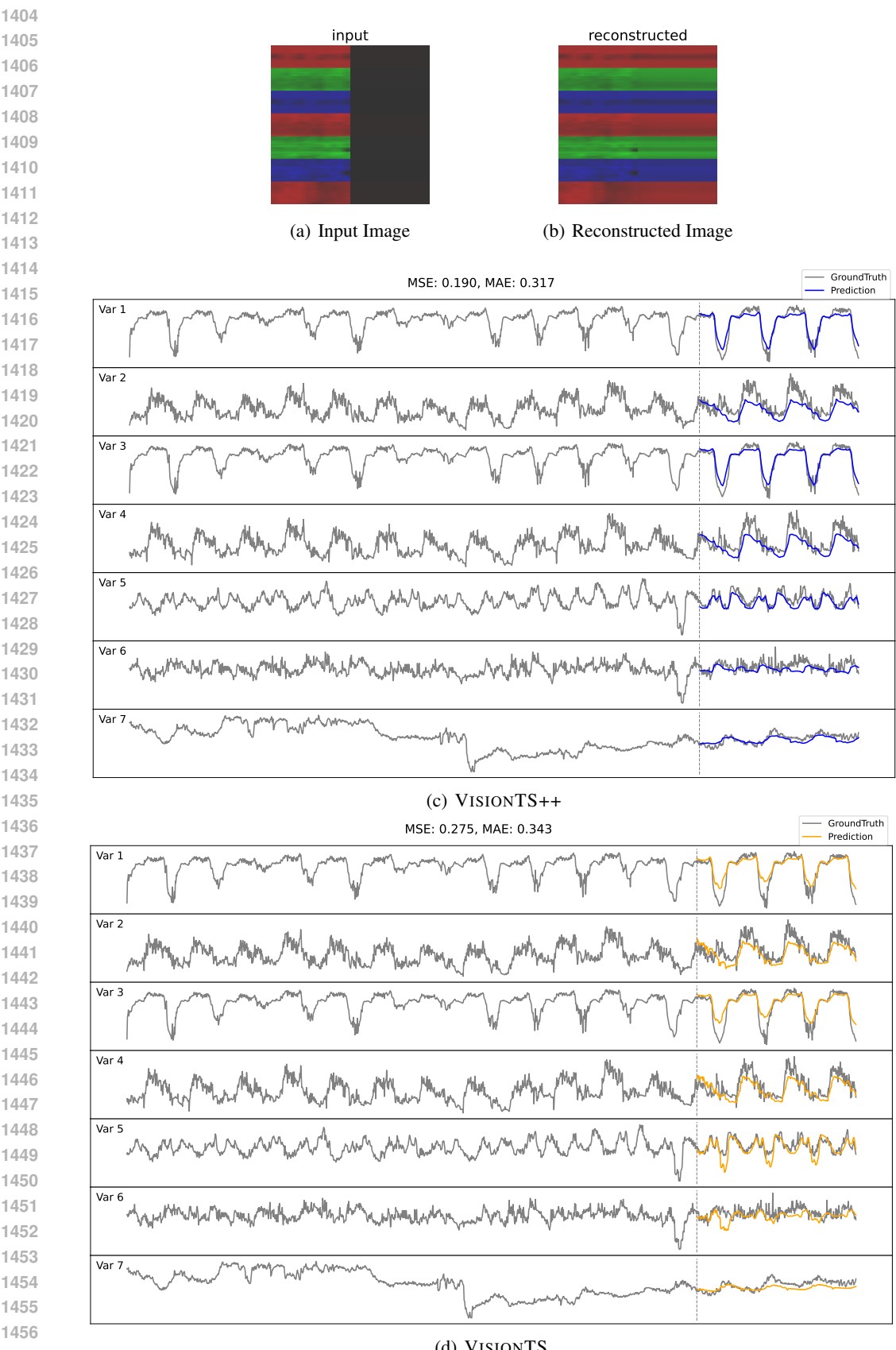

Figure 7: Forecasting visualization on a sample from ETTm1. (a-b) Input/Output images of VI-SIONTS++. (c-d) Prediction comparison between VISIONTS++ and VISIONTS.

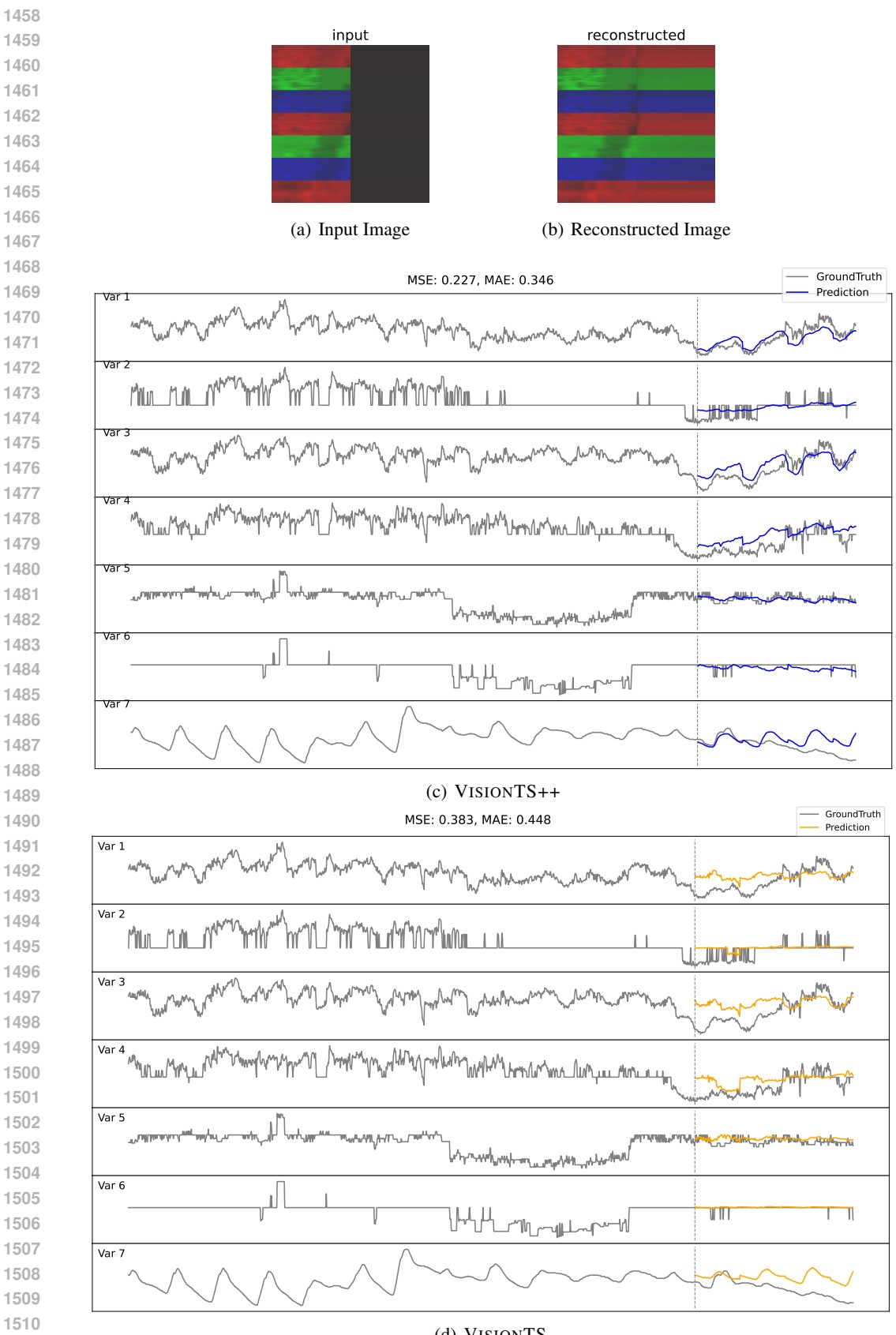

Figure 8: Forecasting visualization on a sample from ETTm2. (a-b) Input/reconstructed images of VISIONTS++. (c-d) Prediction comparison between VISIONTS++ and VISIONTS.

