# OpenReview forum: "VisionTS++: Cross-Modal Time Series Foundation Model with Continual Pre-trained Vision Backbones"
_ICLR.cc/2026/Conference — ICLR 2026 Conference Withdrawn Submission_

### Official Review · Reviewer_r4WL · 2025-10-23

**Soundness:** 4
**Presentation:** 4
**Contribution:** 3
**Rating:** 8
**Confidence:** 3

**Summary:**

This work explores the application potential of vision pre-trained model as time series foundation model. Addressing the challenges in transferring vision models to time series tasks—including 1) the data-modality gap between image data and time series, 2) the multivariate-forecasting gap between RGB channels in vision models and time series structure, and 3) the probabilistic-forecasting gap between vision models and time series predictions—this research proposes the ​VisionTS++​​ model. By leveraging vision models, VisionTS++ represents multivariate time series as multiple RGB sub-images to enhance cross-variable dependency modeling and identify salient temporal features. Additionally, it incorporates multi-quantile probabilistic forecasting. This work further investigates the effectiveness of adapting vision models as temporal foundation models.

**Strengths:**

1.The proposed multi-quantile forecasting provides a solution for converting images into probabilistic predictions that reflect the uncertainty in time series forecasting.

2.The "Vision-Model-Based Filtering" applied to time series appears to be a reasonable and necessary processing step, ensuring that the unbounded nature of time series values can adapt to the bounded pattern of raw image pixels.
3.Extensive experiments on multiple datasets (including the Monash Benchmark, LTSF, and probabilistic forecasting datasets) and competitive results in both in-distribution forecasting and out-of-distribution forecasting validate the model's strong performance.

4.The paper is well-written, complete, and easy to follow.

**Weaknesses:**

1.While the Color-as-boundary strategy appears reasonable, it may exhibit inherent limitations when the number of time series variables significantly exceeds the available RGB channels: On one hand, the intrinsic properties of RGB channels may introduce implicit color bias, causing the model to overemphasize variables associated with certain channels and impair the balance of feature extraction. On the other hand, color reuse may lead to erroneous associations between non-adjacent variables, resulting in the model capturing false dependencies and interfering with the modeling of genuine variable relationships. Moreover, the randomness in color assignment lacks consideration of variable semantic correlations, which can easily lead to scenarios where highly correlated variables are assigned vastly different colors or weakly correlated variables share similar colors, thereby undermining the model's stability and robustness in high-dimensional settings.

2.Although the "Vision-Model-Based Filtering" serves as a reasonable truncation method to reconcile the numerical characteristics of time series with the input compatibility of vision models, it remains questionable whether such normalization adequately accounts for the diversity of time series distributions. For instance, in domains such as healthcare and finance, time series may contain extreme values of significant importance or exhibit upward or downward trends. It is unclear whether the filtering approach remains compatible in such cases. Additionally, other filtering methods worthy of consideration could be explored to accommodate a broader range of time series types.

3.While VisionTS++ explores more suitable input representations for time series data and improvements to the vision model's output head, the MAE backbone remains largely unmodified. Investigating the potential of more diverse vision backbones for time series applications may help enhance forecasting performance. Furthermore, incorporating comparisons with recent vision-based related work [1] would strengthen the rigor and comprehensiveness of the study.

[1] Time-VLM: Exploring Multimodal Vision-Language Models for Augmented Time Series Forecasting.  ICML2025

**Questions:**

In Weakness.

---

> ### Author Response · Authors · 2025-11-26
> **Official Comment by Authors - Part 01**
>
> We sincerely thank the reviewer for recognizing the novelty of our multi-quantile forecasting and vision-model-based filtering, as well as our strong performance across benchmarks. We now carefully address your questions regarding color strategy, filtering robustness, and additional comparisons below.
>
> > **W1: Concerns on Color-as-Boundary Strategy (Bias, Reuse, and Semantic Mismatch)**
>
> We appreciate your detailed analysis of potential biases in the colorization strategy. We clarify that our design explicitly prevents these issues through **randomization** and **attention mechanisms**.
>
> * **Orthogonality to Semantics (Addressing Bias & Reuse):**
>   During pre-training, we randomly shuffle both the input variate order and the RGB color assignment. This ensures that the model learns **color assignments are orthogonal to variate semantics**. In other words, the model learns to use color changes merely as "edge detectors" to distinguish subfigures, rather than associating specific colors with specific variate types or weights.
>
>   * **Empirical Verification:** We conducted a robustness test during inference (**Table 13, Appendix E.3**), comparing (1) Random coloring vs. (2) Fixed cyclic coloring. The results show negligible performance differences, confirming that the model has not learned any implicit color bias or erroneous associations.
>
> * **Capturing True Dependencies (Addressing Semantic Mismatch):**
>   Our model does not rely on "color similarity" to find correlations. Instead, it relies on factual **structural alignment**.
>
>   *   **Evidence:** As visualized in the **Attention Maps (Figure 6, Appendix F.1)**, the model successfully attends to tokens of highly correlated variates at the same timestamp, even when they are assigned totally different colors or are spatially non-adjacent. This proves the model captures genuine dependency based on temporal alignment, unaffected by the color scheme.
>
> For your convenience, we also present Table 13 here.
> | Dataset     | Metric | sequential color | random color |
> | ----------- | ------ | ---------------- | ------------ |
> | Monash      | MAE    | **0.553**        | 0.555        |
> | PF          | MASE   | **0.677**        | **0.677**    |
> |             | CRPS   | **0.515**        | 0.516        |
> | ETTm1       | MSE    | **0.360**        | 0.361        |
> |             | MAE    | **0.372**        | **0.372**    |
> | ETTm2       | MSE    | **0.244**        | **0.244**    |
> |             | MAE    | **0.298**        | **0.298**    |
> | ETTh1       | MSE    | 0.402            | **0.401**    |
> |             | MAE    | 0.416            | **0.415**    |
> | ETTh2       | MSE    | **0.333**        | 0.334        |
> |             | MAE    | **0.370**        | 0.372        |
> | Electricity | MSE    | **0.184**        | **0.184**    |
> |             | MAE    | **0.265**        | **0.265**    |
> | Weather     | MSE    | **0.222**        | 0.223        |
> |             | MAE    | **0.241**        | **0.241**    |
>
> > **W2: Robustness of Vision-Model-Based Filtering (Trends & Extremes)**
>
> We acknowledge the validity of your concern regarding extreme values. Our filtering is a **strategic trade-off** designed for pre-training stability, not a limitation of the model's capability.
>
> * **Strategic Trade-off for Pre-training:**
>   Statistically, extreme outliers (e.g., sensor errors like -9999) are rare but can severely disrupt the distribution learning of a vision backbone pre-trained on natural images.
>
>   *   **Global Gain:** As shown in the ablation study (**Table 7, Appendix D.3**), removing this filtering leads to a clear performance drop. Prioritizing high-quality, in-distribution data for CPT allows the model to learn more robust general patterns, which outweighs the loss of a small fraction of extreme samples.
>
> * **Comparison with Other Filtering Methods:**
>   To explore broader strategies, we compared our method against **Interquartile Range (IQR) [2]** and **Bayesian Online Changepoint Detection (BOCD) [3]** in **Table 16 (Appendix E.6)**.
>   Our vision-based filtering consistently outperforms these statistical methods. This is because our filter is specifically aligned with the **vision model's own constraints** (pixel dynamic range), ensuring the input remains compatible with the backbone's pre-trained knowledge.
>
> * **Inference Handling:** It is important to note that during inference, we **do not discard samples** with trends or extremes. We simply **clip them to the valid range**. This ensures the model can still generate predictions for diverse time series types in finance and healthcare.

---

> ### Author Response · Authors · 2025-11-26
> **Official Comment by Authors - Part 02**
>
> Table 16 for Weakness 2:
> | Dataset     | Metric | Vision-Model-Based | IQN   | BOCD  |
> | ----------- | ------ | ------------------ | ----- | ----- |
> | Monash      | MAE    | **0.553**          | 0.571 | 0.588 |
> | PF          | MASE   | **0.677**          | 0.686 | 0.699 |
> |             | CRPS   | **0.515**          | 0.527 | 0.541 |
> | ETTm1       | MSE    | **0.360**          | 0.378 | 0.392 |
> |             | MAE    | **0.372**          | 0.390 | 0.402 |
> | ETTm2       | MSE    | **0.244**          | 0.256 | 0.273 |
> |             | MAE    | **0.298**          | 0.312 | 0.325 |
> | ETTh1       | MSE    | **0.402**          | 0.409 | 0.422 |
> |             | MAE    | **0.416**          | 0.421 | 0.438 |
> | ETTh2       | MSE    | **0.333**          | 0.336 | 0.336 |
> |             | MAE    | **0.370**          | 0.372 | 0.371 |
> | Electricity | MSE    | **0.184**          | 0.187 | 0.189 |
> |             | MAE    | **0.265**          | 0.268 | 0.275 |
> | Weather     | MSE    | **0.222**          | 0.224 | 0.229 |
> |             | MAE    | **0.241**          | 0.243 | 0.249 |
>
> > **W3: Diverse Vision Backbones and Baselines (Time-VLM)**
>
> We have incorporated the suggested comparisons to strengthen the paper's comprehensiveness.
>
> * **Comparison with Time-VLM [3]:**
>   We added a comparison with **Time-VLM** (ICML 2025) on the LTSF benchmark in **Table 14 (Appendix E.4)**. Our **VisionTS++ (Zero-shot)** consistently outperforms Time-VLM across all metrics, even though Time-VLM operates in a **Few-shot** setting (training on task-specific examples). This highlights the superior generalization capability of our approach.
>
> For your convenience, we also present the simplified Table 14 here.
> | Method      |      | VisionTS++_l |           |      | Time-VLM (few-shot) |       |
> | ----------- | ---- | ------------ | --------- | ---- | ------------------- | ----- |
> | Metric      |      | MSE          | MAE       |      | MSE                 | MAE   |
> | ETTm1       | avg  | **0.354**    | **0.369** |      | 0.364               | 0.385 |
> | ETTm2       | avg  | **0.244**    | **0.299** |      | 0.263               | 0.323 |
> | ETTh1       | avg  | **0.403**    | **0.419** |      | -                   | -     |
> | ETTh2       | vg  | **0.327**    | **0.367** |      | -                   | -     |
> | Electricity | avg  | **0.181**    | **0.264** |      | 0.248               | 0.300 |
> | Weather     | avg  | **0.226**    | **0.244** |      | 0.246               | 0.284 |
> | Average     |      | **0.289**    | **0.327** |      | -                   | -     |
>
>
> * **Alternative Vision Backbones:**
>   We extended our evaluation to include **SimMIM [4]** and **BootMAE [5]** as backbones (**Table 17, Appendix E.7**). While the MAE backbone performs best (likely due to its pixel-reconstruction objective aligning with the forecasting task), models built on SimMIM and BootMAE also outperform several strong baselines (e.g., VisionTS, Moirai). This demonstrates that our **VisionTS++ framework is generalizable** and effective across different vision architectures.
>
> For your convenience, we also present Table 17 here.
> | Dataset     | Metric | MAE       | SimMIM | BootMAE |
> | ----------- | ------ | --------- | ------ | ------- |
> | Monash      | MAE    | **0.553** | 0.605  | 0.579   |
> | PF          | MASE   | **0.677** | 0.719  | 0.693   |
> |             | CRPS   | **0.515** | 0.573  | 0.535   |
> | ETTm1       | MSE    | **0.360** | 0.401  | 0.380   |
> |             | MAE    | **0.372** | 0.403  | 0.393   |
> | ETTm2       | MSE    | **0.244** | 0.278  | 0.259   |
> |             | MAE    | **0.298** | 0.323  | 0.315   |
> | ETTh1       | MSE    | **0.402** | 0.428  | 0.413   |
> |             | MAE    | **0.416** | 0.443  | 0.424   |
> | ETTh2       | MSE    | **0.333** | 0.356  | 0.349   |
> |             | MAE    | **0.370** | 0.387  | 0.381   |
> | Electricity | MSE    | **0.184** | 0.205  | 0.192   |
> |             | MAE    | **0.265** | 0.288  | 0.279   |
> | Weather     | MSE    | **0.222** | 0.239  | 0.229   |
> |             | MAE    | **0.241** | 0.260  | 0.250   |
>
> We hope these responses reinforce your positive assessment of our work. Thank you again for your constructive review!
>
> [1] Detection of outliers using interquartile range technique from intrusion dataset. FICTA 2018.
>
> [2] Bayesian online changepoint detection. Arxiv 2007.
>
> [3] Time-VLM: Exploring Multimodal Vision-Language Models for Augmented Time Series Forecasting. ICML 2025.
>
> [4] Simmim: A simple framework for masked image modeling. CVPR 2022.
>
> [5] Bootstrapped masked autoencoders for vision bert pretraining. ECCV 2022.

---

### Official Review · Reviewer_GUdJ · 2025-10-24

**Soundness:** 3
**Presentation:** 2
**Contribution:** 2
**Rating:** 2
**Confidence:** 4

**Summary:**

This paper proposes VisionTS++, a new time-series foundation model that adapts pretrained vision backbones through continual pre-training on large-scale time-series data. The authors identify three critical challenges—data-modality gap, multivariate-forecasting gap, and probabilistic-forecasting gap—when transferring vision models to time-series forecasting. VisionTS++ addresses these issues via three key components: Vision-Model-Based Filtering, Colorized Multivariate Conversion, and Multi-Quantile Forecasting. The proposed method achieves strong results on 23 datasets across multiple benchmarks, showing consistent improvement over other TSFMs.

**Strengths:**

1. The paper clearly identifies major limitations when applying vision models to time-series forecasting and systematically attempts to address them.
2. Incorporating probabilistic forecasting into the framework is novel and refreshing, extending beyond conventional deterministic designs.

**Weaknesses:**

1. The study only evaluates MAE-based VisionTS++, without validating other vision backbones(SimMIM, BootMAE, etc.). This limits the generality of the proposed framework.
2. The paper does not discuss the computational cost of continual pre-training, raising concerns about training efficiency and scalability.

**Questions:**

1. VisionTS++ appears to use longer or variable look-back windows compared to baselines. Does this lead to unfair comparisons? Moreover, how does the model’s performance vary with respect to input window length?

2. The Vision-Model-Based Filtering step discards samples incompatible with vision model constraints. Does this imply that VisionTS++ cannot handle such series at all? The normalization parameter γ seems manually set—could adjusting it recover those filtered samples?

3. The Colorized Multivariate Conversion introduces several potential issues:

   a) The artificial spatial adjacency between variables might induce bias, as neighboring pixels are assumed to be correlated in image models.

   b) The color assignment scheme lacks theoretical justification and could introduce additional bias.

   c) For high-dimensional series (M > 224), some variables cannot be represented within a fixed image size. How is this issue handled?

4. Using multiple heads in Multi-Quantile Forecasting could exacerbate the computational burden of VisionTS++. Has the efficiency–accuracy trade-off been analyzed?

5. As shown in Figure 2 and Appendix E, the model seems to rely heavily on periodic horizontal replication patterns. If the image conversion were not aligned with the true periodicity, would VisionTS++ still perform effectively?

---

> ### Author Response · Authors · 2025-11-26
> **Official Comment by Authors - Part 01**
>
> **Response to Reviewer 3**
>
> We thank the reviewer for identifying the novelty of our framework and systematic approach to addressing modality gaps. We appreciate your rigorous questions, and we address your concerns with detailed evidence to demonstrate the robustness and correctness of our work.
>
> > **W1: Generality across Vision Backbones (Weakness 1)**
>
> We acknowledge the reviewer for pointing out the importance of verifying whether our framework is specific to MAE or generalizable.
>
> * **New Experiments:** We also implement our framework using **SimMIM** [1] and **BootMAE** [2] as backbones. The results in **Table 17 (Appendix E.7)** show that while they perform slightly worse than MAE, **models built on both SimMIM and BootMAE still outperform several strong TSFM baselines** (e.g., VisionTS and Moirai-Large). This underscores that the effectiveness stems from our **Continual Pre-training (CPT) framework** and the three targeted innovations (filtering, conversion, multi-quantile), not solely from the choice of MAE.
>
> For your convenience in reading, we also present Table 17 here.
>
> | Dataset     | Metric | MAE       | SimMIM | BootMAE |
> | ----------- | ------ | --------- | ------ | ------- |
> | Monash      | MAE    | **0.553** | 0.605  | 0.579   |
> | PF          | MASE   | **0.677** | 0.719  | 0.693   |
> |             | CRPS   | **0.515** | 0.573  | 0.535   |
> | ETTm1       | MSE    | **0.360** | 0.401  | 0.380   |
> |             | MAE    | **0.372** | 0.403  | 0.393   |
> | ETTm2       | MSE    | **0.244** | 0.278  | 0.259   |
> |             | MAE    | **0.298** | 0.323  | 0.315   |
> | ETTh1       | MSE    | **0.402** | 0.428  | 0.413   |
> |             | MAE    | **0.416** | 0.443  | 0.424   |
> | ETTh2       | MSE    | **0.333** | 0.356  | 0.349   |
> |             | MAE    | **0.370** | 0.387  | 0.381   |
> | Electricity | MSE    | **0.184** | 0.205  | 0.192   |
> |             | MAE    | **0.265** | 0.288  | 0.279   |
> | Weather     | MSE    | **0.222** | 0.239  | 0.229   |
> |             | MAE    | **0.241** | 0.260  | 0.250   |
>
> * **Theoretical Justification for MAE:** Furthermore, the superiority of MAE is expected and aligns with recent literature. For instance, **Zhao et al. (2025)** [3] note that SimMIM’s window-based local attention assumes translation invariance—an assumption invalid for imaged time series where spatial position encodes temporal order. Similarly, **Shen et al. (2025)** [4] note that MAE’s ViT-based reconstruction decoder is better suited for pixel-level forecasting tasks than SimMIM’s linear decoder. As for BootMAE, it introduces a momentum encoder (an exponential moving average of the main MAE encoder) may over-smooth high-frequency periodic patterns. Thus, while our framework is general, MAE is currently the optimal choice for pixel-level forecasting tasks.
>
> > **W2: Computational Cost of Pre-training (Weakness 2)**
>
> We added quantitative analysis of training efficiency to address your concern.
>
> *   **3x Efficiency Gain:** Compared to **Moirai**, which requires 1000 iterations (with 512 batch size on NVIDIA A800s) to converge, VisionTS++ achieves superior performance on Monash, PF, and LTSF benchmarks in **less than 300 iterations**. This represents a **>3x efficiency gain**.
> *   **Reason:** This efficiency is directly attributable to our use of the **ImageNet-pretrained MAE backbone**. The learned visual representations provide a high-quality initialization, significantly accelerating the adaptation to time series data compared to training from scratch (as done in Moirai).

---

> ### Author Response · Authors · 2025-11-26
> **Official Comment by Authors - Part 02**
>
> > **Q1: Look-back Window Length: Fairness and Performance (Question 1)**
>
> * **Fairness of Comparison:** Our look-back windows are **not** unfairly long. In fact, they are often shorter than baselines. For example, on ETTh1/h2, **Moirai uses lengths of 5000 and 3000**, whereas we use 2880 and 1728. Meanwhile, GPT4TS [5] also argues that considering the context length as a tunable hyperparameter is reasonable (Appendix H.10, Page 32 in their paper.)
>
> * **Efficiency Advantage:** Crucially, using long windows is not a disadvantage but a strength of our design. While most Transformer-based TSFMs scale quadratically ($O(L^2)$) with look-back window length $L$, VisionTS++ encodes arbitrary lengths into a fixed-size image, achieving **$O(1)$ inference latency** (see **Table 3 in Section 4.4**).
>
>   * Meanwhile, this strategy mirrors recent NLP advances like **DeepSeek-OCR [6]**, which demonstrates that **encoding long text sequences into compact visual tokens** achieves significant compression (up to 10× compression) while retaining high decoding precision (over 96%). This corroborates our core perspective that visual representations serve as highly efficient, high-density carriers for sequential data (time series, language...).
>
> * **Task Difficulty:** From another perspective, **it is challenging for a model to predict a sequence with a length of up to 720 when the input window is very short** (e.g., 60), especially in a zero-shot setting. Therefore, VisionTS++ requires a longer input window to capture the patterns in the input, enabling accurate long-term forecasting.
>
> * **Performance Analysis:** We also added **Figure 5 (Section 4.4)** to analyze sensitivity.
>
>   1.  Our model’s performance **typically improves as the input window length $L$ increases**, but shows **a slight degradation when $L$ becomes too large**—consistent with findings from other TSFMs like Moirai and VisionTS.
>   2.  Initially, a longer window provides **richer historical context**, leading to better performance.
>   3.  However, excessively large window length introduces two issues: (1) spatial constraints in a fixed-width image representation **force interpolation that compresses or discards information**, and (2) longer sequences **may incorporate irrelevant or noisy context**, slightly harming performance.
>
> For your convenience in reading, we also present Table 3 here.
>
> | Context Length    |      | 1k    |      |      |      |      | 1k    | 2k   | 3k   | 4k   |
> | ----------------- | ---- | ----- | ---- | ---- | ---- | ---- | ----- | ---- | ---- | ---- |
> | Prediction Length |      | 1k    | 2k   | 3k   | 4k   |      | 1k    |      |      |      |
> | PatchTST          |      | 0.01  | 0.01 | 0.01 | 0.01 |      | 0.01  | 0.02 | 0.03 | 0.04 |
> | DeepAR            |      | 0.26  | 0.32 | 0.37 | 0.43 |      | 0.26  | 4.06 | 6.10 | 8.17 |
> | LLMTime (8B)      |      | > 200 |      |      |      |      | > 200 |      |      |      |
> | Moirai_base       |      | 0.03  | 0.04 | 0.04 | 0.05 |      | 0.03  | 0.04 | 0.05 | 0.06 |
> | TimesFM           |      | 0.08  | 0.14 | 0.20 | 0.27 |      | 0.07  | 0.13 | 0.20 | 0.25 |
> | VisionTS          |      | 0.04  | 0.03 | 0.03 | 0.03 |      | 0.04  | 0.04 | 0.05 | 0.05 |
> | **VisionTS++**    |      | 0.03  | 0.03 | 0.04 | 0.04 |      | 0.04  | 0.05 | 0.05 | 0.05 |
>
> > **Q2: Handling Filtered Samples & Normalization Parameter $\gamma$ (Question 2)**
>
> * **Handling Strategy (Train vs. Inference):** The "Filtering" is strictly a **training-time** strategy to protect the vision backbone from extreme outliers that disrupt the visual distribution. During **inference**, we do not discard samples; we simply **clip** values to the model's valid pixel range. This ensures robust handling of all series (e.g., there are outliers like -9999 in *Weather* dataset, and these will be clipped).
>
> * **Parameter $\gamma$:** We set $\gamma = 0.4$ based on sensitivity analyses from prior work (VisionTS). It acts like a **physical constant** representing the optimal scaling ratio between two modalities.
>
> * **Alternative Filtering Methods:** As shown in **Table 16 (Appendix E.6)**, our vision-based filtering **outperforms other statistical methods** (IQR [7], BOCD [8]), proving it correctly identifies data incompatible with the specific constraints of visual pre-training.

---

> ### Author Response · Authors · 2025-11-26
> **Official Comment by Authors - Part 03**
>
> Table 16 for Question 2:
> | Dataset     | Metric | Vision-Model-Based | IQN   | BOCD  |
> | ----------- | ------ | ------------------ | ----- | ----- |
> | Monash      | MAE    | **0.553**          | 0.571 | 0.588 |
> | PF          | MASE   | **0.677**          | 0.686 | 0.699 |
> |             | CRPS   | **0.515**          | 0.527 | 0.541 |
> | ETTm1       | MSE    | **0.360**          | 0.378 | 0.392 |
> |             | MAE    | **0.372**          | 0.390 | 0.402 |
> | ETTm2       | MSE    | **0.244**          | 0.256 | 0.273 |
> |             | MAE    | **0.298**          | 0.312 | 0.325 |
> | ETTh1       | MSE    | **0.402**          | 0.409 | 0.422 |
> |             | MAE    | **0.416**          | 0.421 | 0.438 |
> | ETTh2       | MSE    | **0.333**          | 0.336 | 0.336 |
> |             | MAE    | **0.370**          | 0.372 | 0.371 |
> | Electricity | MSE    | **0.184**          | 0.187 | 0.189 |
> |             | MAE    | **0.265**          | 0.268 | 0.275 |
> | Weather     | MSE    | **0.222**          | 0.224 | 0.229 |
> |             | MAE    | **0.241**          | 0.243 | 0.249 |
>
> > **Q3: Potential Issues with Colorized Multivariate Conversion (Question 3)**
>
> * **a) Spatial Bias:** Vision models are not limited to adjacent correlations; they are designed for multi-object analysis. Our **Attention Maps (Figure 6, Appendix F.1)** provide explicit evidence: the model successfully attends to tokens of different variates at the *same timestamp*, even when they are **spatially non-adjacent**.
>
> * **b) Color Bias:** We explicitly mitigate the the spurious correlation by **randomizing** variate ordering and color assignment during training. This ensures the model learns that **color is orthogonal to variate semantics**, using it only as a structural separator. New experiments (**Table 13, Appendix E.3**) confirm that inference performance is invariant to specific color schemes.
>
> * **c) High Dimensions:**
>
>   *   **Channel Grouping:** For datasets with many variates, we employ a **Grouping Strategy**. This is because we limit each image to a maximum of 16 variates to ensure sufficient pixel resolution during CPT process. Therefore, for inference, group size should align with this. For example, on the Weather dataset (21 variates), we split inputs into groups (e.g., 16 + 5), process two images in parallel, and concatenate reconstructed time series results.
>       *   Notably, this grouping strategy relates to the classic trade-off between **channel independence** (group size = 1) and **channel dependence** (larger groups). Both have merit depending on inter-variable relationships.
>       *   Moreover,  conducting time series clustering in advance, aggregating highly correlated variables into the same group may further improve performance. (Reference: Channel Clustering [9]).
>   *   **Larger Backbone:** Simultaneously, adopting vision backbones that support larger input resolutions would naturally raise this variable limit. This direction can be explored in future work.
>
> For your convenience, we also present Table 13 here.
> | Dataset     | Metric | sequential color | random color |
> | ----------- | ------ | ---------------- | ------------ |
> | Monash      | MAE    | **0.553**        | 0.555        |
> | PF          | MASE   | **0.677**        | **0.677**    |
> |             | CRPS   | **0.515**        | 0.516        |
> | ETTm1       | MSE    | **0.360**        | 0.361        |
> |             | MAE    | **0.372**        | **0.372**    |
> | ETTm2       | MSE    | **0.244**        | **0.244**    |
> |             | MAE    | **0.298**        | **0.298**    |
> | ETTh1       | MSE    | 0.402            | **0.401**    |
> |             | MAE    | 0.416            | **0.415**    |
> | ETTh2       | MSE    | **0.333**        | 0.334        |
> |             | MAE    | **0.370**        | 0.372        |
> | Electricity | MSE    | **0.184**        | **0.184**    |
> |             | MAE    | **0.265**        | **0.265**    |
> | Weather     | MSE    | **0.222**        | 0.223        |
> |             | MAE    | **0.241**        | **0.241**    |

---

> ### Author Response · Authors · 2025-11-26
> **Official Comment by Authors - Part 04**
>
> > **Q4: Computational Burden of Multi-Quantile Heads (Question 4)**
>
> We perform a detailed analysis in **Table 10 & 11 (Appendix E.1)** to prove the cost is negligible.
>
> * **Minimal Parameters and Inference Latency:** Adding quantile heads is extremely cheap. The 9-head variant adds only **3.0M parameters** to the 112M backbone. The calculation is: $8 \text{ extra heads} \times 512 \text{ dim} \times (16 \times 16 \times 3) \approx 3.0\text{M}$. And this **incurs almost no additional latency penalty during inference** (see results in Table 11).
>
> * **Forecasting Performance:** However, based on Table 10, adding heads **greatly improves performance**, especially for probabilistic forecasting (CRPS metric in PF benchmark).
>
> For your convenience, we also present the tables here.
>
> Table 10:
> | Dataset     | Metric | 9-heads   | 3-heads | 1-head    |
> | ----------- | ------ | --------- | ------- | --------- |
> | Monash      | MAE    | **0.553** | 0.561   | 0.570     |
> | PF          | MASE   | **0.677** | 0.697   | 0.721     |
> |             | CRPS   | **0.515** | 0.675   | 0.804     |
> | ETTm1       | MSE    | **0.360** | 0.362   | 0.364     |
> |             | MAE    | **0.372** | 0.375   | 0.374     |
> | ETTm2       | MSE    | **0.244** | 0.249   | 0.251     |
> |             | MAE    | **0.298** | 0.301   | 0.302     |
> | ETTh1       | MSE    | 0.402     | 0.404   | **0.398** |
> |             | MAE    | 0.416     | 0.416   | **0.413** |
> | ETTh2       | MSE    | **0.333** | 0.336   | 0.340     |
> |             | MAE    | **0.370** | 0.375   | 0.372     |
> | Electricity | MSE    | **0.184** | 0.192   | 0.198     |
> |             | MAE    | **0.265** | 0.279   | 0.285     |
> | Weather     | MSE    | **0.222** | 0.227   | 0.228     |
> |             | MAE    | **0.241** | 0.247   | 0.248     |
>
> Table 11:
> | Context Length       |      | 1k   |      |      |      |      | 1k   | 2k   | 3k   | 4k   |
> | -------------------- | ---- | ---- | ---- | ---- | ---- | ---- | ---- | ---- | ---- | ---- |
> | Prediction Length    |      | 1k   | 2k   | 3k   | 4k   |      | 1k   |      |      |      |
> | VisionTS             |      | 0.04 | 0.03 | 0.03 | 0.03 |      | 0.04 | 0.04 | 0.05 | 0.05 |
> | VisionTS++ (1-head)  |      | 0.03 | 0.03 | 0.03 | 0.03 |      | 0.04 | 0.04 | 0.05 | 0.05 |
> | VisionTS++ (3-heads) |      | 0.03 | 0.03 | 0.03 | 0.04 |      | 0.04 | 0.05 | 0.05 | 0.05 |
> | VisionTS++ (9-heads) |      | 0.03 | 0.03 | 0.04 | 0.04 |      | 0.04 | 0.05 | 0.05 | 0.05 |

---

> ### Author Response · Authors · 2025-11-26
> **Official Comment by Authors - Part 05**
>
> > **Q5: Dependence on Periodicity Alignment (Question 5)**
>
> * **Practical Determination:** In real-world applications, periodicity $P$ is reliably derived from sampling frequency (e.g., hourly=24) or Fast Fourier Transform (FFT), so misalignment is rare.
>
> * **Robustness Test ($P=1$):** To address your concern, we also test the model by forcing $P=1$ on all LTSF datasets with clear periodicity (**Table 15, Appendix E.5**). While performance drops compared to using the true periodicity, **VisionTS++ with $P=1$ still achieves competitive results** (comparable to Moirai-Large).
>
> * **Why it works:** Even without periodic stacking, the horizontal axis preserves the **temporal trend**. Furthermore, our CPT exposes the model to diverse aperiodic data, enabling it to generalize even when explicit periodic structures are absent.
>
> For your convenience, we also present Table 15 here.
> | Dataset     | Metric | VisionTS++ | VisionTS++ & P=1 | Moirai_large |
> | ----------- | ------ | ---------- | ---------------- | ------------ |
> | ETTm1       | MSE    | **0.360**  | 0.448            | 0.390        |
> |             | MAE    | **0.372**  | 0.406            | 0.389        |
> | ETTm2       | MSE    | **0.244**  | 0.296            | 0.276        |
> |             | MAE    | **0.298**  | 0.333            | 0.320        |
> | ETTh1       | MSE    | **0.402**  | 0.438            | 0.510        |
> |             | MAE    | **0.416**  | 0.444            | 0.469        |
> | ETTh2       | MSE    | **0.333**  | 0.399            | 0.354        |
> |             | MAE    | **0.370**  | 0.411            | 0.377        |
> | Electricity | MSE    | **0.184**  | 0.202            | 0.188        |
> |             | MAE    | **0.265**  | 0.291            | 0.273        |
> | Weather     | MSE    | **0.222**  | 0.244            | 0.260        |
> |             | MAE    | **0.241**  | 0.266            | 0.275        |
> | Average     | MSE    | **0.291**  | 0.338            | 0.330        |
> |             | MAE    | **0.327**  | 0.359            | 0.351        |
>
> We hope these detailed clarifications and additional experiments demonstrate that VisionTS++ is a robust, efficient, and methodologically sound foundation model.
>
> [1] Simmim: A simple framework for masked image modeling. CVPR 2022.
>
> [2] Bootstrapped masked autoencoders for vision bert pretraining. ECCV 2022.
>
> [3] From Images to Signals: Are Large Vision Models Useful for Time Series Analysis? Arxiv 2025.
>
> [4] Multi-Modal View Enhanced Large Vision Models for Long-Term Time Series Forecasting. Arxiv 2025.
>
> [5] One fits all: Power general time series analysis by pretrained LM. Neurips 2023.
>
> [6] DeepSeek-OCR: Contexts Optical Compression. Arxiv 2025.
>
> [7] Detection of outliers using interquartile range technique from intrusion dataset. FICTA 2018.
>
> [8] Bayesian online changepoint detection. Arxiv 2007.
>
> [9] From Similarity to Superiority: Channel Clustering for Time Series Forecasting. Neurips 2024.

---

> > ### Comment · Reviewer_GUdJ · 2025-11-26
> >
> > Thank you for the comprehensive response. The additional experiments using SimMIM and BootMAE as backbones, as well as the analysis of efficiency, have alleviated part of my initial concerns, and I have accordingly increased my evaluation of the paper.
> >
> > However, I believe several key issues still remain and require further clarification from the authors:
> >
> > Q1: Fairness Regarding Look-back Window Length
> >
> > 1. The authors note that Moirai uses a longer look-back window. However, many baseline models in the paper (e.g., PatchTST, TiDE) were originally configured with an input window of 512, and this was not standardized in the experiments, potentially affecting fairness.
> > 2. More seriously, the authors should be aware that these datasets are constructed using sliding-window slicing. Changing the look-back window directly alters the training and testing samples (both in count and content), leading to an even more significant fairness issue in comparison.
> >
> > Q3: Potential Issues with Colorized Multivariate Conversion
> >
> > 1. Spatial Bias: The response suggests that global attention enables capturing non-adjacent dependencies, which I understand. However, this does not rule out ordering-induced bias. I recommend adding an experiment where the same 16-variable series is fed into the model under multiple different variable permutations. If the results remain consistent, I would consider this sufficient evidence that ordering does not introduce noticeable bias.
> > 2. High-Dimensional Variables: The proposed variable grouping/selection strategy remains unclear for high-dimensional settings (e.g., >16 variables). Relying on random selection would be concerning. The authors should clarify how variable grouping is determined in such cases.
> >
> > Q5: Dependence on Periodicity Alignment
> >
> > 1. The authors state that the periodicity $P$ is derived from the sampling rate or the Fourier transform.
> >
> >    I would like to know: When using FFT, is $P$ computed on the entire dataset, or individually for each input segment? To my knowledge, FFT-derived periods on single segments are often unstable.
> >
> > 2. The additional experiment sets $P = 1$, which effectively removes periodicity.
> >
> >    I am more interested in how the model behaves under period misalignment, for example: The true period is $P = 24$, but the model is given $P = 18$. How sensitive is the method to such misalignment?
> >
> > If the authors can clarify the above points, I will have greater confidence in the rigor and contribution of this work.

---

> > > ### Author Response · Authors · 2025-11-28
> > > **Official Response to New Comments by Reviewer GUdJ - Part 01**
> > >
> > > We sincerely thank the reviewer for the continued engagement and for recognizing the value of our additional experiments (SimMIM/BootMAE backbones, efficiency analysis). We appreciate the opportunity to clarify the remaining concerns regarding fairness, bias, and periodicity.
> > >
> > > **Q1: Fairness Regarding Look-back Window Length**
> > >
> > > **1. Clarification on Test Sample Count (Q1.2):**
> > > We respectfully clarify a potential misunderstanding regarding the sliding window mechanism. We follow the standard implementation from **[TS-Library]** (a commonly-used codebase for many TSF models) [1].
> > >
> > > *   **Mechanism:** Let's take "Dataset_Custom" class as an example, where "border1s = [0, num_train - seq_len, len(df_raw) - num_test - seq_len]; border2s = [num_train, num_train + num_vali, len(df_raw)]; border1 = border1s[set_type]; border2 = border2s[set_type]". And the final returned sample number is: "len(data_x) - seq_len - pred_len + 1". This means that in the evaluation/test phase, the number of test samples is calculated as: `len(df_raw) - (len(df_raw) - num_test - seq_len) - seq_len - pred_len + 1 = num_test - pred_len + 1`. And the "num_test" is only influenced by train/val/test split proportion, e.g., under 7:1:2 split, `num_test =  int(len(df_raw) * 0.2)`.
> > > *   **Conclusion:** This ensures that once the setting is fixed (pred_len, train/val/test split proportion), the **total number of test samples is constant and independent of the look-back window length**. The only factor that changes is the training set size (which slightly decreases as look-back increases), but since VisionTS++ operates in a **zero-shot** setting without training, this does not affect us. Thus, the comparison is mathematically fair.
> > >
> > > **2. Impact of Look-back Length (Q1.1):**
> > > We reiterate that optimal performance typically follows an inverted-U curve with respect to look-back length $L$ (see **Figure 5, Section 4.4**). Baselines like PatchTST/TiDE use relatively shorter windows (e.g., 512) may due to their empirical optimum; longer inputs may conversely degrade their performance due to overfitting or noise. Meanwhile, to fully address your concern, we also conducted new experiments as follows.
> > >
> > > *   **New Experiment (VisionTS++ with L=512):** we evaluated VisionTS++ with a standardized look-back of **$L=512$** on both PF and LTSF benchmarks.
> > >     *   **PF Benchmark:** As shown in the table below, reducing $L$ to 512 even improves CRPS on part of datasets, compared to $L=2000$. While the average performance drops slightly, it still significantly outperforms full-shot baselines (PatchTST, TiDE, which we even tuned over $L \in [48, 2880]$ follow Moirai's setting) and the Moirai-large model.
> > >     *   **LTSF Benchmark:** Similarly, with $L=512$, VisionTS++ sees a minor performance drop but still beats TFSMs such as VisionTS, Time-MoE-base, and Chronos-large on average.
> > >     *   **Result Summary:** This confirms that our superior performance stems from the model architecture, not just the long look-back window.
> > >
> > > PF:
> > > | |Zero-shot||| |Full-shot||
> > > |---|---|---|---|---|---|---|
> > > |Dataset|Method|VisionTS++_b(L=512)|VisionTS++_b(L=2000)|Moirai_l|PatchTST|TiDE|
> > > |Electricity|CRPS|0.045|**0.042**|0.050|0.052|0.048|
> > > ||MASE|0.655|**0.631**|0.751|0.753|0.706|
> > > |Solar|CRPS|**0.334**|0.355|0.406|0.518|0.420|
> > > ||MASE|**1.080**|1.155|1.237|1.607|1.265|
> > > |Walmart|CRPS|**0.064**|**0.064**|0.098|0.082|0.077|
> > > ||MASE|0.702|**0.689**|1.007|0.867|0.814|
> > > |Weather|CRPS|**0.039**|0.044|0.051|0.059|0.054|
> > > ||MASE|0.492|**0.447**|0.515|0.844|0.832|
> > > |Istanbul Traffic|CRPS|**0.104**|0.115|0.112|0.112|0.110|
> > > ||MASE|**0.588**|0.616|0.631|0.653|0.618|
> > > |Turkey Power|CRPS|0.042|**0.036**|0.036|0.054|0.046|
> > > ||MASE|0.784|**0.737**|0.870|1.234|0.904|
> > > |Norm.|CRPS|0.523|**0.515**|0.609|0.679|0.612|
> > > ||MASE|0.689|**0.677**|0.794|0.937|0.827|
> > > |1st count||6|9|1|0|0|
> > >
> > > LTSF:
> > > |Dataset|Method|VisionTS++_b (L=512)|VisionTS++_b|VisionTS|Time-MoE_b|Chronos_l|
> > > |---|---|---|---|---|---|---|
> > > |ETTm1|MSE|0.388|**0.360**|0.374|0.376|0.556|
> > > ||MAE|0.376|**0.372**|0.372|0.406|0.465|
> > > |ETTm2|MSE|0.269|**0.244**|0.282|0.316|0.295|
> > > ||MAE|0.310|**0.298**|0.321|0.361|0.338|
> > > |ETTh1|MSE|0.418|0.402|**0.390**|0.394|0.589|
> > > ||MAE|0.417|0.416|**0.414**|0.420|0.466|
> > > |ETTh2|MSE|0.338|**0.333**|**0.333**|0.405|0.455|
> > > ||MAE|0.372|**0.370**|0.375|0.415|0.427|
> > > |Electricity|MSE|0.199|**0.184**|0.207|-|0.204|
> > > ||MAE|0.270|**0.265**|0.294|-|0.274|
> > > |Weather|MSE|0.230|**0.222**|0.269|0.270|0.279|
> > > ||MAE|0.244|**0.241**|0.292|0.300|0.306|
> > > |Average|MSE|0.307|**0.291**|0.309|-|0.396|
> > > ||MAE|0.332|**0.327**|0.345|-|0.379|

---

> > > ### Author Response · Authors · 2025-11-28
> > > **Official Response to New Comments by Reviewer GUdJ - Part 03**
> > >
> > > Results of different ratios between inference periodicity and true periodicity are provided here. Best results are in bold, and second-best results are in italic.
> > >
> > > |Dataset|Metric|1/4P|1/2P|3/4P|P|5/4P|3/2P|2P|3P|Moirai_large|
> > > |---|---|---|---|---|---|---|---|---|---|---|
> > > |ETTm1|MSE|0.472|0.408|0.513|**0.312**|0.503|0.483|*0.328*|0.362|0.390|
> > > ||MAE|0.408|0.397|0.455|**0.342**|0.450|0.443|*0.361*|0.379|0.389|
> > > |ETTm2|MSE|0.189|0.185|0.204|**0.167**|0.204|0.205|*0.172*|0.183|0.276|
> > > ||MAE|0.273|0.270|0.287|**0.245**|0.288|0.294|*0.252*|0.265|0.320|
> > > |ETTh1|MSE|0.454|0.435|0.493|**0.368**|0.499|0.482|*0.376*|0.385|0.510|
> > > ||MAE|0.426|0.419|0.455|**0.392**|0.470|0.439|*0.401*|0.408|0.469|
> > > |ETTh2|MSE|0.304|0.297|0.308|**0.267**|0.305|0.299|*0.271*|0.276|0.354|
> > > ||MAE|0.358|0.346|0.360|**0.317**|0.362|0.353|*0.323*|0.330|0.377|
> > > |Electricity|MSE|0.199|0.185|0.215|**0.147**|0.220|0.205|*0.158*|0.166|0.188|
> > > ||MAE|0.278|0.269|0.291|**0.233**|0.296|0.284|*0.242*|0.249|0.273|
> > > |Weather|MSE|0.194|0.183|0.210|**0.146**|0.217|0.206|*0.159*|0.169|0.260|
> > > ||MAE|0.211|0.208|0.239|**0.179**|0.252|0.228|*0.188*|0.197|0.275|
> > >
> > > References:
> > >
> > > [1] https://github.com/thuml/Time-Series-Library
> > >
> > > [2] From Similarity to Superiority: Channel Clustering for Time Series Forecasting. Neurips 2024.
> > >
> > > [3] Gluonts: Probabilistic and neural time series modeling in python. JMLR 2020.
> > >
> > > [4] Monash time series forecasting archive. Arxiv 2021.
> > >
> > > [5] Fedformer: Frequency enhanced decomposed transformer for long-term series forecasting. ICML 2022.
> > >
> > > [6] TimesNet: Temporal 2D-Variation Modeling for General Time Series Analysis. ICLR 2023.
> > >
> > > [7] Calibration of time-series forecasting: Detecting and adapting context-driven distribution shift. KDD 2024.

---

> ### Author Response · Authors · 2025-11-28
> **Official Response to New Comments by Reviewer GUdJ - Part 02**
>
> **Q3: Potential Issues with Colorized Multivariate Conversion**
>
> **1. Ordering-Induced Bias (Q3.1):**
> Thanks for your suggestion. As previously mentioned, we **entirely randomized both variate ordering and color assignment** during pre-training. This can theoretically orthogonalize real semantics from variate orders/colors. Meanwhile, we also add new experiments as you mentioned.
>
> *   **New Experiment:** We performed inference on the same multivariate series under different permutations: **Original Order**, **Reversed Order**, and **Random Shuffle**.
> *   **Result:** The metrics remain highly consistent across all permutations (Table below). This provides sufficient evidence that **no variate ordering bias is introduced**, confirming the model treats variates based on their values and temporal alignment, not their sequence index.
>
> |Dataset|Metric|original|reversed|random|
> |---|---|---|---|---|
> |Monash|MAE|**0.553**|**0.553**|0.554|
> |PF|MASE|**0.677**|**0.677**|**0.677**|
> ||CRPS|**0.515**|**0.515**|0.516|
> |ETTm1|MSE|0.360|**0.359**|**0.359**|
> ||MAE|0.372|**0.371**|0.372|
> |ETTm2|MSE|**0.244**|**0.244**|0.245|
> ||MAE|**0.298**|0.299|0.299|
> |ETTh1|MSE|0.402|**0.401**|**0.401**|
> ||MAE|0.416|**0.415**|0.416|
> |ETTh2|MSE|**0.333**|**0.333**|**0.333**|
> ||MAE|**0.370**|0.371|0.371|
> |Electricity|MSE|**0.184**|0.185|0.185|
> ||MAE|**0.265**|0.266|0.266|
> |Weather|MSE|**0.222**|0.223|0.223|
> ||MAE|**0.241**|**0.241**|**0.241**|
>
> **2. High-Dimensional Grouping Strategy (Q3.2):**
> We apologize for the lack of clarity.
>
> *   **Mechanism:**
>     *   Our Grouping Strategy includes a `group_size` parameter, which controls the maximum variates in one image. It is set to 16 during continual pre-training, which aims to ensure sufficient pixel spaces for each variate.
>     *   During inference, `group_size` can also be set as 16 (align with training). If varaite number $M > 16$ (e.g., Weather with $M=21$), we split variates sequentially into groups (e.g., $[1,2,...,16]$ and $[17,18,...,21]$), process them as two separate images in parallel, getting two forecasts, and finally **concatenate** them.
> *   **Justification:** While such sequential splitting can be heuristic, it is effective (as seen in our Weather results in Table 1 in the paper). This can also be understood as a combination thinking of channel dependence and channel independence.
> *   **Future Work:** We agree that advanced grouping, such as clustering highly correlated variables (Reference: *Channel Clustering [2], NeurIPS 2024*), might further improve performance, which we would leave for future exploration.
>
>
> **Q5: Dependence on Periodicity Alignment**
>
> **1. Periodicity Calculation (Q5.1):**
> *   **Method:** In this paper, we primarily use the **sampling frequency** (e.g., hourly=24) as it is stable and standard (also used by: GluonTS [3], Monash [4]).
> *   **FFT Stability:** When frequency metadata is unavailable, we can use FFT.
>     *   *Whole Dataset:* Preferred for offline tasks (like LTSF) where the entire frequency/periodicity can be obtained as prior knowledge, guaranteeing stability.
>     *   *Instance-level:* When the global periodicity is hard to get, instance-level FFT is a also proven technique used in **FEDformer [5], TimesNet [6], SOLID [7], etc.**. These works demonstrate that local period estimation is sufficiently robust.
>     *   *Fallback:* Even if estimation fails, our $P=1$ fallback ensures the model doesn't collapse.
>
> **2. Sensitivity to Misalignment (Q5.2):**
> Thanks for your problem, this is an interesting question. To generalize this problem, we conducted a stress test on the LTSF benchmark by setting the inference period $P_{infer}$ to various ratios of the true period $P_{true}$ ($1/4, 1/2, 3/4, 1, 5/4, 3/2, 2, 3$).
>
> *   **Results:**
>     *   **Robust Cases:** Performance is best at integer multiples ($2P, 3P$), and the unit fractions ($1/2P, 1/4P$), where the structural alignment is still partially preserved.
>     *   **Sensitive Cases:** Performance drops for fractional misalignments like $3/4P$ or $5/4P$, as these cause "phase shifting" that disrupts both inter-period seasonality and entire trend continuity.
>     *   **Conclusion:** Despite the drop, the results in misaligned cases (e.g., on ETTh/ETTm2/Weather) **remain better than Moirai-large**. This indicates that while alignment helps, the model's learned representations are robust enough to handle imperfect settings. Moreover, the $P=1$ fallback always remains a feasible option.
>
> Tables and results are provided in the part-03.

---

### Official Review · Reviewer_qqnm · 2025-11-01

**Soundness:** 3
**Presentation:** 3
**Contribution:** 2
**Rating:** 4
**Confidence:** 4

**Summary:**

To address the data–modality gap, the multivariate–forecasting gap, and the probabilistic–forecasting gap, this paper proposes VisionTS++, a vision-model-based time series foundation model trained through large-scale pretraining. VisionTS++ consists of three key components: vision-model-based filtering, colorized multivariate time series conversion, and multi-quantile forecasting. After pretraining, VisionTS++ achieves strong forecasting performance in both in-distribution and out-of-distribution scenarios.

**Strengths:**

1. The paper proposes a new vision-model-based time series foundation model, which achieves good forecasting performance across multiple benchmarks.
2. The paper clearly identifies the key challenges of applying vision models to time series analysis, including the data–modality gap and the multivariate–forecasting gap.
3. The overall writing of the paper is clear and well-organized.

**Weaknesses:**

1. The paper presents an incremental improvement over VisionTS, with the proposed modules—vision-model-based filtering, colorized multivariate conversion, and multi-quantile forecasting—being relatively straightforward. The filtering module performs simple threshold-based filtering; the multivariate conversion resembles prior vision-based time series models such as ViTST (NeurIPS 2023); and the multi-quantile forecasting capability has already been incorporated in most recent TSFMs.
2. Regarding vision-model-based filtering, the method relies on a simple threshold mechanism, which may inadvertently remove large but legitimate variations (e.g., those caused by major events or holidays), potentially leading to inaccurate predictions in such cases.
3. For multivariate conversion, the method concatenates different time series variables into an image representation. While the vision model may implicitly capture variable dependencies, the approach does not explicitly model variable correlations, which could lead to suboptimal channel correlation modeling. It would be helpful if the authors could provide further evidence or analysis to support the model’s effectiveness in this regard.
4. Table 1 and 2 lacks comparisons with more recent baselines, such as Sundial (ICML 2025).
5. The paper also lacks an efficiency analysis compared with other TSFMs.

**Questions:**

See Weaknesses.

---

> ### Author Response · Authors · 2025-11-26
> **Official Comment by Authors - Part 01**
>
> We thank the reviewer for recognizing the potential of our vision-model-based TSFM and good forecasting performance. We also appreciate the opportunity to clarify our work and provide additional evidence to address your concerns.
>
> > **W1: Novelty and Comparison with Prior Works (Weakness 1)**
>
> Firstly, we respectfully clarify that **VisionTS++ is not merely an incremental update to VisionTS**, but a fundamental shift from *format alignment* to **semantic alignment**.
> While VisionTS primarily aligns time series and images in *format* to explore potential, VisionTS++ aims to **bridge the semantic modality gap** through **Continual Pre-training (CPT)** on large-scale time series data. However, as noted in the Introduction (Lines 78-85), directly applying CPT for transferring knowledge is non-trivial and prone to **negative transfer**, due to three fundamental gaps (**Data-Modality, Multivariate, and Probabilistic Gaps**) between the vision modality and time series modality.
> This challenge mirrors findings in language-to-time-series transfer; for instance, Time-LLM [1] has to reprogram input time series into text prototype representations to align with the LLM's inherent capabilities. Similarly, specific adaptations or designs are **indispensable** for our framework to mitigate these cross-modal gaps effectively.
> We then give clarifications on each of our approaches:
>
> *   **Vision-Model-Based Filtering:** This is not just a statistical threshold but an **Out-of-Distribution (OOD) detector for the vision backbone**. Standard outliers in time series might be valid, but values exceeding the pixel dynamic range break the pre-trained visual representation space. Filtering them is essential to protect the model's original visual knowledge during CPT. We also conduct **comparison to other filtering approaches** in the following section for addressing your concerns on Weakness 2.
> *   **Contrast with ViTST (NeurIPS 2023) [2]:** We fundamentally differ from ViTST in information density, alignment and scalability:
>     *   **Information Density:** ViTST uses line plots, which result in significant **pixel redundancy (white space)**. Our periodic transformation + vertical stacking creates a dense 2D matrix with maximal information per pixel.
>     *   **Temporal Alignment:** ViTST’s line graphs place variates at arbitrary pixel coordinates, making temporal alignment difficult. Our **vertical stacking explicitly aligns timestamps** across variates (e.g., time series at time $t$ for all variates are in the same column in the image), enabling the model to capture cross-variate dependencies via spatial attention.
>     *   **Scalability:** ViTST relies on variate-specific colors, which is hard to scale. Our approach only requires distinct colors for adjacent subfigures, making it scalable to arbitrary variate numbers.
> *   **Multi-Quantile Forecasting:** This design is motivated by **alignment with the MAE's native pre-train task**. That is, we repurpose the MAE’s original ability to "predict image pixels" to "predict multiple image pixels". This is a more natural extension of the visual backbone, and also aligns with MAE's original capability and knowledge. However, most TSFMs are empirical choices for architectural changes, without clear motivation for task alignment.
>
> > **W2: Vision-Model-Based Filtering and "Legitimate Variations" (Weakness 2)**
>
> We acknowledge the concern about removing legitimate events (e.g., holidays). However, our design is a **strategic trade-off for pre-training robustness**:
>
> 1. **Global Performance:** As shown in the ablation study (**Table 7, Appendix D.3**), removing this filtering leads to a performance drop. Statistically, extreme outliers are rare, but they disproportionately harm the stability of visual pre-training. Prioritizing high-quality, in-distribution data for CPT yields better generalization and robustness. That is, we sacrifice performance on a small number of extreme-value time series to achieve better prediction on the majority of normal ones and better robustness.
>
> 2. **Superiority over Statistical Filters:** To validate our approach, we also compared it against standard statistical filters: **Interquartile Range (IQR)** [3] and **Bayesian Online Changepoint Detection (BOCD)** [4]. As reported in **Table 16 (Appendix E.6)**, our vision-model-based filtering consistently outperforms these complex methods. This confirms that filtering based on the **vision model's own constraints** (pixel boundaries) is more effective for this specific architecture than generic statistical filtering.

---

> ### Author Response · Authors · 2025-11-26
> **Official Comment by Authors - Part 02**
>
> Table 16 for Weakness 2:
>  | Dataset     | Metric | Vision-Model-Based | IQN   | BOCD  |
>  | ----------- | ------ | ------------------ | ----- | ----- |
>  | Monash      | MAE    | **0.553**          | 0.571 | 0.588 |
>  | PF          | MASE   | **0.677**          | 0.686 | 0.699 |
>  |             | CRPS   | **0.515**          | 0.527 | 0.541 |
>  | ETTm1       | MSE    | **0.360**          | 0.378 | 0.392 |
>  |             | MAE    | **0.372**          | 0.390 | 0.402 |
>  | ETTm2       | MSE    | **0.244**          | 0.256 | 0.273 |
>  |             | MAE    | **0.298**          | 0.312 | 0.325 |
>  | ETTh1       | MSE    | **0.402**          | 0.409 | 0.422 |
>  |             | MAE    | **0.416**          | 0.421 | 0.438 |
>  | ETTh2       | MSE    | **0.333**          | 0.336 | 0.336 |
>  |             | MAE    | **0.370**          | 0.372 | 0.371 |
>  | Electricity | MSE    | **0.184**          | 0.187 | 0.189 |
>  |             | MAE    | **0.265**          | 0.268 | 0.275 |
>  | Weather     | MSE    | **0.222**          | 0.224 | 0.229 |
>  |             | MAE    | **0.241**          | 0.243 | 0.249 |
>
> > **W3: Explicit vs. Implicit Multivariate Modeling (Weakness 3)**
>
> We argue that our Colorized Multivariate Conversion enables **explicit** correlation modeling through structural alignment. By stacking variates vertically, we force the model to process them jointly. To prove the model captures these correlations, we visualized the **Attention Maps (Figure 6, Appendix F.1)**.
>
> *   **Result:** The attention maps show that when querying a token from one variate, the model assigns high attention scores to tokens of *other* variates at the **same temporal index**, even when they are spatially non-adjacent.
> *   **Conclusion:** This confirms that after CPT, the model explicitly learns to look for cross-variate dependencies, leveraging the spatial alignment we designed.
>
> > **W4: Comparison with Sundial (Weakness 4)**
>
> We have added comprehensive comparisons with Sundial (ICML 2025) [5]:
>
> 1.  **GIFT-Eval:** Figure 4 includes the ranking of `Sundial-Base-128M`. Our VisionTS++ (both Base and Large) achieves a better average rank.
> 2.  **LTSF Benchmark:** We added a zero-shot comparison with `Sundial-Large` in **Table 14 (Appendix E.4)**. VisionTS++ consistently outperforms Sundial across MSE and MAE metrics.
> 3.  **Data Efficiency:** Notably, Sundial is pre-trained on *TimeBench*, which fully subsumes our training set (LOTSA) and is significantly larger (LOTSA is only ~22% of TimeBench). VisionTS++ achieves superior performance with much less pre-training data, highlighting our data efficiency.
>
> + For your convenience, we also present simplified Table 14 here.
>
> | Method      |      | VisionTS++_l |           |      | Sundial_large |           |
> | ----------- | ---- | ------------ | --------- | ---- | ------------- | --------- |
> | Metric      |      | MSE          | MAE       |      | MSE           | MAE       |
> | ETTm1       | avg  | 0.354        | **0.369** |      | **0.331**     | **0.369** |
> | ETTm2       | avg  | **0.244**    | **0.299** |      | 0.254         | 0.315     |
> | ETTh1       | avg  | 0.403        | **0.419** |      | **0.395**     | 0.420     |
> | ETTh2       | avg  | **0.327**    | **0.367** |      | 0.337         | 0.387     |
> | Electricity | avg  | 0.181        | 0.264     |      | **0.166**     | **0.262** |
> | Weather     | avg  | **0.226**    | **0.244** |      | 0.238         | 0.275     |
> | Average     |      | **0.289**    | **0.327** |      | 0.291         | 0.338     |

---

> ### Author Response · Authors · 2025-11-26
> **Official Comment by Authors - Part 03**
>
> > **W5: Efficiency Analysis (Weakness 5)**
>
> We have included a detailed efficiency analysis in **Table 3 (Section 4.4)**.
>
> * **Inference Speed:** VisionTS++ has comparable speed to VisionTS and Moirai. The multi-quantile design is highly efficient: adding 8 extra heads to the Base model only introduces **3.0M parameters** (negligible compared to the 112M backbone), incurring almost no latency penalty, while significantly improves performance, especially for probabilistic forecasting.
>
> * **Complexity:** Crucially, while most Transformer-based TSFMs typically scale with $O(L^2)$ regarding look-back length $L$, VisionTS++ maintains **nearly constant latency**. This is because we encode variable-length inputs into a fixed-size image representation, achieving **$O(1)$ inference efficiency** regarding the sequence length.
>
>  For your convenience in reading, we also present Table 3 here.
>
> | Context Length    |      | 1k    |      |      |      |      | 1k    | 2k   | 3k   | 4k   |
> | ----------------- | ---- | ----- | ---- | ---- | ---- | ---- | ----- | ---- | ---- | ---- |
> | Prediction Length |      | 1k    | 2k   | 3k   | 4k   |      | 1k    |      |      |      |
> | PatchTST          |      | 0.01  | 0.01 | 0.01 | 0.01 |      | 0.01  | 0.02 | 0.03 | 0.04 |
> | DeepAR            |      | 0.26  | 0.32 | 0.37 | 0.43 |      | 0.26  | 4.06 | 6.10 | 8.17 |
> | LLMTime (8B)      |      | > 200 |      |      |      |      | > 200 |      |      |      |
> | Moirai_base       |      | 0.03  | 0.04 | 0.04 | 0.05 |      | 0.03  | 0.04 | 0.05 | 0.06 |
> | TimesFM           |      | 0.08  | 0.14 | 0.20 | 0.27 |      | 0.07  | 0.13 | 0.20 | 0.25 |
> | VisionTS          |      | 0.04  | 0.03 | 0.03 | 0.03 |      | 0.04  | 0.04 | 0.05 | 0.05 |
> | **VisionTS++**    |      | 0.03  | 0.03 | 0.04 | 0.04 |      | 0.04  | 0.05 | 0.05 | 0.05 |
>
>
>
> We hope these responses adequately address your concerns. Thank you for your valuable suggestions!
>
> References:
>
> [1] Time-LLM: Time Series Forecasting by Reprogramming Large Language Models. ICLR2024.
> [2] Time series as images: Vision transformer for irregularly sampled time series. Neurips 2023.
>
> [3] Detection of outliers using interquartile range technique from intrusion dataset. FICTA 2018.
>
> [4] Bayesian online changepoint detection. Arxiv 2007.
>
> [5] Sundial: A Family of Highly Capable Time Series Foundation Models. ICML 2025.

---

### Official Review · Reviewer_PTdf · 2025-11-02

**Soundness:** 2
**Presentation:** 2
**Contribution:** 2
**Rating:** 4
**Confidence:** 3

**Summary:**

VisionTS++ adapts a Masked Auto-Encoder (MAE) pre-trained on ImageNet to universal time-series forecasting via three tweaks: (i) vision-based filtering that discards sequences whose normalised values fall outside the pixel range expected by the MAE, (ii) “colourised” multi-subfigure images that stack an arbitrary number of variates into one 3-channel picture, and (iii) some parallel reconstruction heads trained with quantile loss to deliver probabilistic forecasts without parametric assumptions. After continual pre-training on 231 B points (LOTSA) the model is evaluated on Monash (29 data-sets), LTSF, PF and GIFT-Eval benchmarks, beating recent TSFMs by 6–44 % MSE and ranking first on most tasks.

**Strengths:**

(1) first paper that systematically closes the data-range, multivariate and probabilistic gaps when turning an off-the-shelf vision backbone into a competitive TSFM.
(2) no new attention layers, no patch re-design; only lightweight heads and input/output converters—easy to reproduce.
(3) SOTA on 4 widely used benchmarks (31/62 first places on LTSF, best nMAE on Monash, top CRPS on PF, 1st on GIFT-Eval) with both base and large variants.
(4) removing filtering (−7 %), colourisation (−12 %) or multi-quantile heads (−10 %) consistently hurts, validating each gimmick.

**Weaknesses:**

(1) the core idea (TS ➔ image ➔ MAE) is identical; improvements come from three engineering accessories rather than a new modelling principle.
(2) no analysis of why pixel-range filtering or random RGB boundaries should be optimal; no guarantee that vision inductive biases align with temporal dynamics.
(3) only forecasting; classification, anomaly detection or irregular sampling not tested.
(4) no study on (i) #quantile heads h, (ii) alternative change-point or range-based filters, (iii) image size W or subfigure layout, (iv) datasets outside LOTSA/GIFT.
(5) several typos (“appedix”, “quarntie”), some tables missing std-dev, captions duplicated.

**Questions:**

See Weaknesses.

---

> ### Author Response · Authors · 2025-11-26
> **Official Comment by Authors - Part 01**
>
> We thank the reviewer for the constructive feedback and for recognizing our work’s strengths, particularly its **SOTA performance across widely used benchmarks**. We now carefully address your concerns as follows.
>
> >  **W1: Novelty of the core idea and "Engineering Accessories"**
>
> We respectfully clarify that **VisionTS++ is fundamentally distinct from VisionTS in both goal and methodology**, and our contributions are **methodological innovations essential for cross-modal alignment**, not merely engineering accessories.
>
> *   **From Format to Semantic Alignment:** While VisionTS primarily aligns time series and images in *format* to explore potential, VisionTS++ aims to **bridge the semantic modality gap** through **Continual Pre-training (CPT)** on large-scale time series data. However, as noted in the Introduction (Lines 78-85), directly applying CPT for transferring knowledge is non-trivial and prone to **negative transfer**, due to three fundamental gaps  (**Data-Modality, Multivariate, and Probabilistic Gaps**) between the vision modality and time series modality.
>     This challenge mirrors findings in language-to-time-series transfer; for instance, Time-LLM [1] has to reprogram input time series into text prototype representations to align with the LLM's inherent capabilities. Similarly, specific adaptations or designs are **indispensable** for our framework to mitigate these cross-modal gaps effectively.
> *   **Essential Solutions, Not Accessories:** The three proposed components are meticulously designed to solve the above obstacles and modality gaps that make direct continual pre-training suboptimal:
>     1.  **Vision-Model-Based Filtering:** Solves the *Data-Modality Gap* by preventing low-quality data (incompatible with the vision backbone's distribution) from corrupting pre-trained knowledge—a problem inherent to CPT that VisionTS does not face.
>     2.  **Colorized Multivariate Conversion:** Solves the *Multivariate Gap* by enabling fixed 3-channel vision models to model high-dimensional dependencies via "multi-object analysis" capability.
>     3.  **Multi-Quantile Forecasting:** Solves the *Probabilistic Gap*, equipping deterministic reconstruction models with uncertainty quantification capabilities without parametric assumptions.
>
>
> > **W2.1: No analysis of why pixel-range filtering should be optimal & W4-(ii): alternative change-point or range-based filters**
>
> Thanks for your valuable suggestions for pointing out this. We have added extensive analyses and experiments to validate our design choices.
>
> * We compared our "Vision-Model-Based Filtering" against two standard statistical filtering strategies: (1) **Interquartile Range (IQR)** [2] based outlier detection, and (2) **Bayesian Online Changepoint Detection (BOCD)** [3].
>
> * As shown in **Table 16 (Appendix E.6)**, both alternatives yield consistently worse results. Standard filters discard statistical outliers that might actually be informative, whereas our method specifically filters samples that are **"out-of-distribution" for the pre-trained vision backbone**. This ensures the model receives data that aligns with its visual inductive biases, achieving a robust trade-off between data quality and quantity.
>
>   * For your convenience in reading, we also present Table 16 here.
>
>   * | Dataset     | Metric | Vision-Model-Based | IQN   | BOCD  |
>     | ----------- | ------ | ------------------ | ----- | ----- |
>     | Monash      | MAE    | **0.553**          | 0.571 | 0.588 |
>     | PF          | MASE   | **0.677**          | 0.686 | 0.699 |
>     |             | CRPS   | **0.515**          | 0.527 | 0.541 |
>     | ETTm1       | MSE    | **0.360**          | 0.378 | 0.392 |
>     |             | MAE    | **0.372**          | 0.390 | 0.402 |
>     | ETTm2       | MSE    | **0.244**          | 0.256 | 0.273 |
>     |             | MAE    | **0.298**          | 0.312 | 0.325 |
>     | ETTh1       | MSE    | **0.402**          | 0.409 | 0.422 |
>     |             | MAE    | **0.416**          | 0.421 | 0.438 |
>     | ETTh2       | MSE    | **0.333**          | 0.336 | 0.336 |
>     |             | MAE    | **0.370**          | 0.372 | 0.371 |
>     | Electricity | MSE    | **0.184**          | 0.187 | 0.189 |
>     |             | MAE    | **0.265**          | 0.268 | 0.275 |
>     | Weather     | MSE    | **0.222**          | 0.224 | 0.229 |
>     |             | MAE    | **0.241**          | 0.243 | 0.249 |

---

> ### Author Response · Authors · 2025-11-26
> **Official Comment by Authors - Part 02**
>
> > **W2.2 No analysis of why random RGB boundaries should be optimal**
>
> * **Rationale for Random RGB Boundaries:**
>
>   * **Visual Inductive Bias:** Boundaries are necessary to treat variates as distinct "objects." Using RGB channels leverages the vision model's inherent ability to distinguish objects via color as edges.
>
>   * **Orthogonality to Semantics:** During training, we entirely randomize variate-to-color mapping. This ensures the model learns that **color assignments are orthogonal to variate semantics**, preventing spurious correlations.
>
>   * **Robustness Verification:** We conducted new inference experiments comparing (1) Random coloring vs. (2) Fixed cyclic coloring (Red-Green-Blue). **Table 13 (Appendix E.3)** shows negligible performance difference, proving the model has learned to use color solely as a structural separator, independent of the specific color value.
>
>     * For your convenience in reading, we also present Table 13 here.
>
>     * | Dataset     | Metric   | sequential color | random color |
>       | ----------- | -------- | ---------------- | ------------ |
>       | Monash      | MAE      | **0.553**        | 0.555        |
>       | PF          | MASEtabl | **0.677**        | **0.677**    |
>       |             | CRPS     | **0.515**        | 0.516        |
>       | ETTm1       | MSE      | **0.360**        | 0.361        |
>       |             | MAE      | **0.372**        | **0.372**    |
>       | ETTm2       | MSE      | **0.244**        | **0.244**    |
>       |             | MAE      | **0.298**        | **0.298**    |
>       | ETTh1       | MSE      | 0.402            | **0.401**    |
>       |             | MAE      | 0.416            | **0.415**    |
>       | ETTh2       | MSE      | **0.333**        | 0.334        |
>       |             | MAE      | **0.370**        | 0.372        |
>       | Electricity | MSE      | **0.184**        | **0.184**    |
>       |             | MAE      | **0.265**        | **0.265**    |
>       | Weather     | MSE      | **0.222**        | 0.223        |
>       |             | MAE      | **0.241**        | **0.241**    |
>
> > **W2.3: No guarantee that vision inductive biases align with temporal dynamics**
>
> We argue that visual biases naturally align with temporal properties through our design:
>
> *   **Locality & Periodicity:** Vision models excel at **spatial locality**. Our transformation maps time series periodicity to spatial blocks, preserving both intra-period locality (vertical) and inter-period dependencies (horizontal), which aligns with the temporal-locality and seasonality of time series dynamics.
> *   **Cross-Variate Dependency:** As shown in the **Attention Maps (Figure 6, Appendix F.1)**, the model successfully attends to tokens of different variates at the same timestamp, even when they are spatially non-adjacent. This confirms that our *Colorized Multivariate Conversion* successfully repurposes the "multi-object analysis" capability of vision models for multivariate time series modeling.
> *   **Supported by Recent Literature:** The compatibility between vision priors and temporal dynamics is further validated by a growing body of work. **ViTST** [4] demonstrates that the spatial attention in ViTs is effective for temporal dependency modeling. Similarly, **ViTime** [5] and **ImagenTime** [6] show that transforming time series into visual representations (e.g., line plots or STFT maps) allows vision models to excel at forecasting and generation tasks. These studies collectively confirm that visual inductive biases are robustly transferable to time series analysis when the data is appropriately structured.
>
> > **W3: Scope of Tasks**
>
> While we agree that classification and anomaly detection are interesting, **focusing on forecasting is consistent with the standard for recent top-tier Time Series Foundation Models (TSFMs)**, such as *Moirai [7], Chronos [8], TimesFM [9], and Sundial [10]*, which primarily target forecasting. So our work should be sufficient to support the content of a conference paper.
> However, VisionTS++ also has the potential for these tasks (e.g., using the [CLS] token for classification or image reconstruction for imputation), which we have noted in the last paragraph in the *Conclusion and Future Work* section.

---

> ### Author Response · Authors · 2025-11-26
> **Official Comment by Authors - Part 03**
>
> > **W4: Additional Experiments and Ablations**
> We have included new experiments in the Appendix to address your questions:
> * **Quantile Head Number ($h$):** We test $h=\{1, 3, 9\}$ in **Table 10 (Appendix E.1)**. Results show that increasing $h$ consistently improves performance, especially CRPS, with negligible parameter increase (only +3.0M params for 9 heads).
>
>   * For your convenience in reading, we also present the Table 10 here.
>
>   * | Dataset     | Metric | 9-heads   | 3-heads | 1-head    |
>     | ----------- | ------ | --------- | ------- | --------- |
>     | Monash      | MAE    | **0.553** | 0.561   | 0.570     |
>     | PF          | MASE   | **0.677** | 0.697   | 0.721     |
>     |             | CRPS   | **0.515** | 0.675   | 0.804     |
>     | ETTm1       | MSE    | **0.360** | 0.362   | 0.364     |
>     |             | MAE    | **0.372** | 0.375   | 0.374     |
>     | ETTm2       | MSE    | **0.244** | 0.249   | 0.251     |
>     |             | MAE    | **0.298** | 0.301   | 0.302     |
>     | ETTh1       | MSE    | 0.402     | 0.404   | **0.398** |
>     |             | MAE    | 0.416     | 0.416   | **0.413** |
>     | ETTh2       | MSE    | **0.333** | 0.336   | 0.340     |
>     |             | MAE    | **0.370** | 0.375   | 0.372     |
>     | Electricity | MSE    | **0.184** | 0.192   | 0.198     |
>     |             | MAE    | **0.265** | 0.279   | 0.285     |
>     | Weather     | MSE    | **0.222** | 0.227   | 0.228     |
>     |             | MAE    | **0.241** | 0.247   | 0.248     |
>
> * **Alternative Change-point or Range-based Filters:** See above contents for Weakness 2.1.
>
> * **Subfigure Layout:** We compared "Left-to-Right" with "Top-to-Bottom" and "Right-to-Left" layouts in **Table 12 (Appendix E.2)**. The performance is almost identical, suggesting the image reconstruction process is **isotropic** and robust to spatial arrangement.
>
>   * For your convenience in reading, we also present Table 12 here.
>
>   * | Dataset     | Metric | left -> right | top -> bottom | right -> left |
>     | ----------- | ------ | ------------- | ------------- | ------------- |
>     | Monash      | MAE    | **0.553**     | 0.555         | 0.554         |
>     | PF          | MASE   | 0.677         | **0.676**     | 0.677         |
>     |             | CRPS   | 0.515         | **0.513**     | 0.516         |
>     | ETTm1       | MSE    | **0.360**     | 0.362         | 0.360         |
>     |             | MAE    | **0.372**     | 0.375         | 0.373         |
>     | ETTm2       | MSE    | **0.244**     | 0.248         | 0.246         |
>     |             | MAE    | **0.298**     | 0.303         | 0.300         |
>     | ETTh1       | MSE    | 0.402         | 0.401         | **0.399**     |
>     |             | MAE    | 0.416         | 0.415         | **0.413**     |
>     | ETTh2       | MSE    | **0.333**     | 0.336         | 0.333         |
>     |             | MAE    | **0.370**     | 0.374         | 0.371         |
>     | Electricity | MSE    | **0.184**     | 0.186         | 0.188         |
>     |             | MAE    | **0.265**     | 0.268         | 0.270         |
>     | Weather     | MSE    | 0.222         | 0.223         | **0.221**     |
>     |             | MAE    | 0.241         | 0.243         | **0.240**     |
>
> * **Image Size:** The size is constrained to $224 \times 224$ by the pre-trained MAE architecture. However, our framework can support scaling to larger vision backbones with larger image input in future work.
>
> * **Datasets:** We have conducted on both **LOTSA** and **GIFT-Eval-Pretrain** datasets. LOTSA [7] contains over 27B observations across nine domains (Energy, Transport, Climate, CloudOps, Web, Sales, Nature, Econ/Fin, and Healthcare); And GIFT-Eval-Pretrain [11] is more diverse and balanced than LOTSA, including BuildingsBench, ClimateLearn, CloudOps TSF, GluonTS, LargeST, LibCity, SubseasonalClimateUSA, ProEnFo, Monash, and LOTSA-Others. So we believe these datasets are sufficiently diverse and large-scale to validate the model's generalization capability on different pre-trained datasets.
>
> > **W5: Typos and Missing Standard Deviations:**
>
> Thanks for pointing these out! We have corrected all typos (e.g., "appendix") and removed duplicate captions. We have also updated **Table 8 and Table 9 in Appendix D.4** to include the **standard deviations** for both LTSF and PF benchmarks, derived from 5 independent runs. The consistently low standard deviations across datasets further demonstrate **the stability of our model and the robustness of our training process**.

---

> ### Author Response · Authors · 2025-11-26
> **Official Comment by Authors - Part 04 (References)**
>
> We hope these responses adequately address your concerns. Thank you for your valuable suggestions.
>
> The following are some references.
>
> [1] Time-LLM: Time Series Forecasting by Reprogramming Large Language Models. ICLR2024.
>
> [2] Detection of outliers using interquartile range technique from intrusion dataset. FICTA 2018.
>
> [3] Bayesian online changepoint detection. Arxiv 2007.
>
> [4] Time series as images: Vision transformer for irregularly sampled time series. Neurips 2023.
>
> [5] Vitime: A visual intelligence-based foundation model for time series forecasting. Arxiv 2024.
>
> [6] Utilizing image transforms and diffusion models for generative modeling of short and long time series. Neurips 2024.
>
> [7] Unified Training of Universal Time Series Forecasting Transformers. ICML 2024.
>
> [8] Chronos: Learning the language of time series. Arxiv 2024.
>
> [9] A decoder-only foundation model for time-series forecasting. ICML 2024.
>
> [10] Sundial: A Family of Highly Capable Time Series Foundation Models. ICML 2025.
>
> [11] GIFT-Eval: A Benchmark for General Time Series Forecasting Model Evaluation. Neurips 2024.

---

### Author Response · Authors · 2025-11-26
**Global comments**

**Global Response to All Reviewers**

We thank the reviewers for their insightful comments and constructive feedback. We are pleased that they generally agree our paper presents a novel and systematic framework for adapting vision backbones to time series forecasting. The paper is recognized for identifying and closing the critical data-modality, multivariate, and probabilistic gaps (R PTdf, R qqnm, R r4WL), and for achieving **state-of-the-art performance** across widely used benchmarks including Monash, LTSF, PF, and GIFT-Eval (R PTdf, R qqnm, R r4WL). The reviewers also appreciate the paper’s novel integration of **multi-quantile forecasting** for uncertainty quantification (R GUdJ, R r4WL) and its clear, reproducible writing (R qqnm, R4). **We believe that our work opens a new road to leveraging continual pre-trained vision models for universal and efficient time series forecasting.**

We value the reviewers' suggestions and have incorporated all feedback into the revised version. Significant updates and new experiments are highlighted in $\color{red}{Red}$ in the revision. The key improvements include:

*   **In-depth Model Analysis (Section 4.4):** We added a computational cost analysis to verify the $O(1)$ inference efficiency of our image-based approach (Table 3) and conducted a sensitivity analysis on look-back window lengths (Figure 5).
*   **Comprehensive New Experiments (Appendix E):** We included extensive supplementary experiments to address specific concerns:
    *   **Ablations & Robustness:** Impact of quantile head numbers (E.1), subfigure layouts (E.2), color assignment strategies (E.3), and periodicity settings ($P=1$) (E.5).
    *   **New Comparisons:** Comparisons with **Sundial** and **Time-VLM** (E.4), alternative filtering strategies (IQR, BOCD) (E.6), and different vision backbones (SimMIM, BootMAE) (E.7).
*   **Statistical Rigor (Appendix D.4):** We updated the LTSF and PF benchmark results to include standard deviations derived from 5 independent runs (Tables 8 & 9).
*   **Visualization of Attention Maps (Appendix F.1):** We added attention map visualizations (Figure 6) to explicitly demonstrate how the model captures cross-variate dependencies.

Next, we will address each reviewer's concerns individually.

---

### Note · Authors · 2026-01-29

I have read and agree with the venue's withdrawal policy on behalf of myself and my co-authors.

---

### Meta-Review · Area_Chair_zx7J · 2026-01-02

**Summary:**

The reviews are predominantly negative, specifically from 2, 3, 4, with the most expressing dissatisfaction in Weakness parts. Consequently, PTdf concers that improvements come from three engineering accessories rather than a new modelling principle. qqnm express concerns about incremental novlety over VisonTS, since the vision-model-based filtering, colorized multivariate conversion, anmulti-quantile forecasting is relatively straightforward;GUdJ show concerns about the design of VisionTS++. All reviewers share common concerns about insufficient experiment comparison like efficiency analysis. Only  r4WL is positive, yet express concerns about Color-as-boundary strategy and Vision-Model-Based Filtering. Althoug the authors have provided detailed responses, only part of concerns have been well solved. Therefore, I persist in recommending the rejection of this paper in its current form.

**Reviewer Concerns:**

r4WL is positive, and the concerns have been solved based on the rebuttal. For others, only part concerns of PTdf, qqnm, GUdJ  have been addressed.

**Reviewer Scores:**

PTdf, and qqnm may keep the original score as 4.
GUdJ expressed that he would increase the evaluation of the paper, thus he may raise the score from 2 to 4.
r4WL is positive, and he may keep the original score as 8.

---

### Decision · Program_Chairs · 2026-01-26

Reject